# A Bootstrap Perspective on Stochastic Gradient Descent

## Abstract

Machine learning models trained with *stochastic* gradient descent (SGD) can generalize better than those trained with deterministic gradient descent (GD). In this work, we study SGD's impact on generalization through the lens of the statistical bootstrap: SGD uses gradient variability under batch sampling as a proxy for solution variability under the randomness of the data collection process. We use empirical results and theoretical analysis to substantiate this claim. In idealized experiments on empirical risk minimization, we show that SGD is drawn to parameter choices that are robust under resampling and thus avoids spurious solutions even if they lie in wider and deeper minima of the training loss. We prove rigorously that by implicitly regularizing the trace of the gradient covariance matrix, SGD controls the algorithmic variability. This regularization leads to solutions that are less sensitive to sampling noise, thereby improving generalization. Numerical experiments on neural network training show that explicitly incorporating the estimate of the algorithmic variability as a regularizer improves test performance. This fact supports our claim that bootstrap estimation underpins SGD's generalization advantages.

## 1 Introduction

### 1.1 Background

Modern machine learning models are typically overparameterized and/or non-convex, resulting in many parameter choices that achieve good training performance. However, the test performance of these parameter choices can be vastly different, making the training algorithm an important element of generalization. Notably, SGD tends to find training loss minima that generalize better on test data than GD (Zhang et al., 2016). This work aims to clarify the mechanism underlying this phenomenon.

Some studies explain this phenomenon by suggesting that the noise in SGD induces it toward flatter minima in the loss landscape, which they argue are associated with better generalization performance (Keskar et al., 2016; Yang et al., 2023; Wu & Su, 2023). However, this explanation is undermined by the lack of invariance under function reparameterization (Dinh et al., 2017; Andriushchenko et al., 2023). Another line of work provides stability-based bounds on the generalization gap (Bousquet et al., 2020; Zhou et al., 2022). These approaches usually assume uniform smoothness of the loss function, which can be overly loose in certain regions for complex loss functions. Consequently, these bounds may be trivial at solutions to which the algorithms converge. To address these limitations, we introduce a bootstrap estimation perspective to understand the generalization advantage of SGD.

### 1.2 Our Contributions

We propose that the mini-batch gradient variability in SGD acts as a bootstrap estimate of the solution's sensitivity to resampling, which we term algorithmic variability, and SGD implicitly regularizes this bootstrap estimate to enhance generalization. This perspective motivates the design of new regularizers that can further improve generalization. Our main contributions are:

- We conduct an idealized experiment of function optimization to show that the gradient variability plays an important role in the generalization performance of SGD. More specifically, the data-dependent gradient noise steers SGD away from regions with high variability.

- Under certain assumptions, we derive an approximation of the expected generalization gap, which is determined by the solution's Hessian matrix and the algorithmic variability with respect to sampling noise. We further derive an approximate upper bound on the algorithmic variability, which consists of two components. We propose that SGD utilizes the accumulated gradient variability as a bootstrap estimate of the first component of the algorithmic variability bound and implicitly regularizes it, thereby enhancing generalization.

- We conduct numerical experiments on SGD with explicit regularizers corresponding to estimates of the two components of the algorithmic variability bound. The results demonstrate that both components are essential for reducing test losses and that regularizers based on these estimates can be effectively applied in neural network training. To the best of our knowledge, no prior work has employed the second component of the algorithmic variability bound as a regularizer.

## 1.3 PAPER OUTLINE

Section 2 introduces the key concepts used in this work and conducts an idealized experiment to illustrate the importance of data-dependent gradient noise in helping SGD generalize better. Section 3 discusses the main theoretical conclusions of this work. We first prove that the expected generalization gap depends on the algorithmic variability. Then, we propose that SGD implicitly regularizes the bootstrap estimate of a bound on the algorithmic variability to enhance generalization. Section 4 provides experimental results that support our analysis and show that estimates of the algorithmic variability bound can be used as explicit regularizers. Section 5 reviews related work. Section 6 concludes this paper.

## 2 PRELIMINARIES

### 2.1 EMPIRICAL RISK MINIMIZATION AND GENERALIZATION GAP

Because the population distribution is inaccessible, the training loss, also called the empirical risk, is minimized as a surrogate for the population loss. The difference between the training loss and the population loss, known as the generalization gap, quantifies how well the model generalizes.

### 2.2 STOCHASTIC GRADIENT DESCENT

Gradient-based methods are widely employed for optimizing objective functions in machine learning. Unlike standard GD, which updates the model parameters with the gradient of the entire training set, SGD uses the gradient of a mini-batch randomly sampled from the training set at each iteration. The sampling noise in SGD can be captured by a gradient noise term in its update rule. Initially introduced to improve scalability with large datasets, SGD has demonstrated superior generalization performance with various models and tasks compared with GD. We will show in Section 2.4 that the data-dependent noise is essential in pushing SGD out of minima that generalize poorly.

### 2.3 BOOTSTRAP ESTIMATION

Given an estimator, we may wish to know how it would have differed over different samples. Bootstrap estimation measures this variability by treating the training set as an empirical distribution and evaluating the variability of the estimate over subsamples drawn from it.

The gradient variability of SGD evaluates how much the gradient changes with different samples from the training set. This connection motivates our explanation of SGD's generalization behavior through the lens of bootstrap estimation.

### 2.4 AN IDEALIZED EXPERIMENT

We use an idealized experiment to show that the sampling noise, or equivalently, the gradient noise, can induce SGD to converge to solutions with better generalization compared to GD. To show the importance of the data-dependent noise, we also conduct experiments with NoisyGD, which adds data-independent Gaussian noise to each GD update.

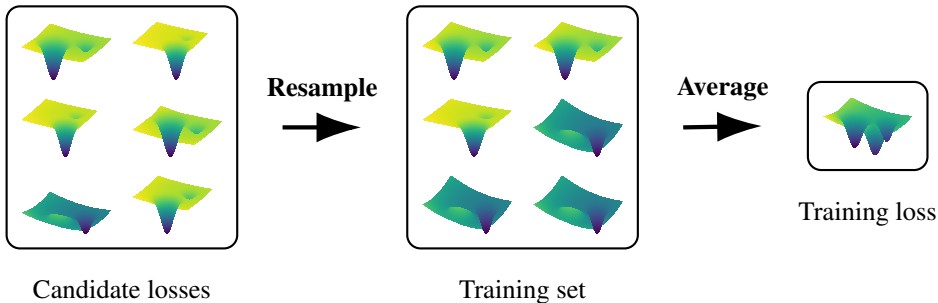

Figure 1: Construction of loss functions in the idealized experiment.

We run GD, SGD, and NoisyGD to optimize two-dimensional objective functions with the same initializations and hyperparameters. Each objective function is constructed by sampling 30 functions with replacement from a set of 30 candidate functions and averaging them, as illustrated in Figure 1. For each algorithm, we report the mean test loss over 100 runs with different initializations. The detailed experimental setup is in Appendix A.1.

Table 1: Average test losses of algorithms in the idealized experiment.

| ALGORITHM | TEST LOSS |
| --- | --- |
| GD | $13.99 \pm 1.67$ |
| SGD | $5.33 \pm 1.73$ |
| NoisyGD | $11.09 \pm 1.71$ |

Random sampling can produce training sets with samples deviating substantially from other samples in the population. These deviations yield spurious minima in the training loss landscape that fit the training data well but generalize poorly. Avoiding such spurious minima is crucial for generalization.

Figure 2 reports the experimental results. The population and training loss heat maps show a better-generalizing minimum at $(7, 7)$ and a spurious minimum at $(7, 1)$. Among the three algorithms, SGD spends most of the training time in regions where the trace of the gradient covariance matrix is small. Even though the spurious minimum is deeper, broader, and has smaller gradient norms in its neighborhood than the better-generalizing minimum in the training loss landscape, SGD still converges to the latter owing to its smaller variability. This observation is corroborated in Table 1, where SGD has the smallest average test loss among the algorithms. These results indicate that the data-dependent gradient noise enables SGD to avoid converging to spurious minima arising from random sampling from the population. This finding inspires the idea that SGD utilizes gradient variability to estimate the sensitivity of the solution to different training data.

## 3 THEORETICAL RESULTS

In Section 2.4, we show with the idealized experiment that SGD converges to solutions with small gradient variability, which generalize well to the test data. This observation raises two key questions: (1) what factor drives SGD to solutions with small gradient variability? (2) how does this reduced gradient variability improve generalization? In this section, we formally establish the connection between the gradient variability and generalization. First, under the assumptions that SGD can achieve small gradient on the training data and that replacing one training sample has only a minor impact on the solution, we show that the expected generalization gap can be decomposed into the trace of the product between the solution's Hessian and the algorithmic variability, which measures the sensitivity of the solution to replacing a single sample in the training set. Then, we demonstrate that the gradient variability of SGD can be regarded as a bootstrap estimate of the first component of a bound on the algorithmic variability. Lastly, we show that the implicit regularizer of SGD, as characterized by Smith et al. (2021), is equivalent to regularizing gradient variability. Taken together, these points suggest that the implicit regularizer steers SGD toward solutions with smaller gradient variability, which, being a bootstrap estimate of the algorithmic variability, leads to improved generalization.

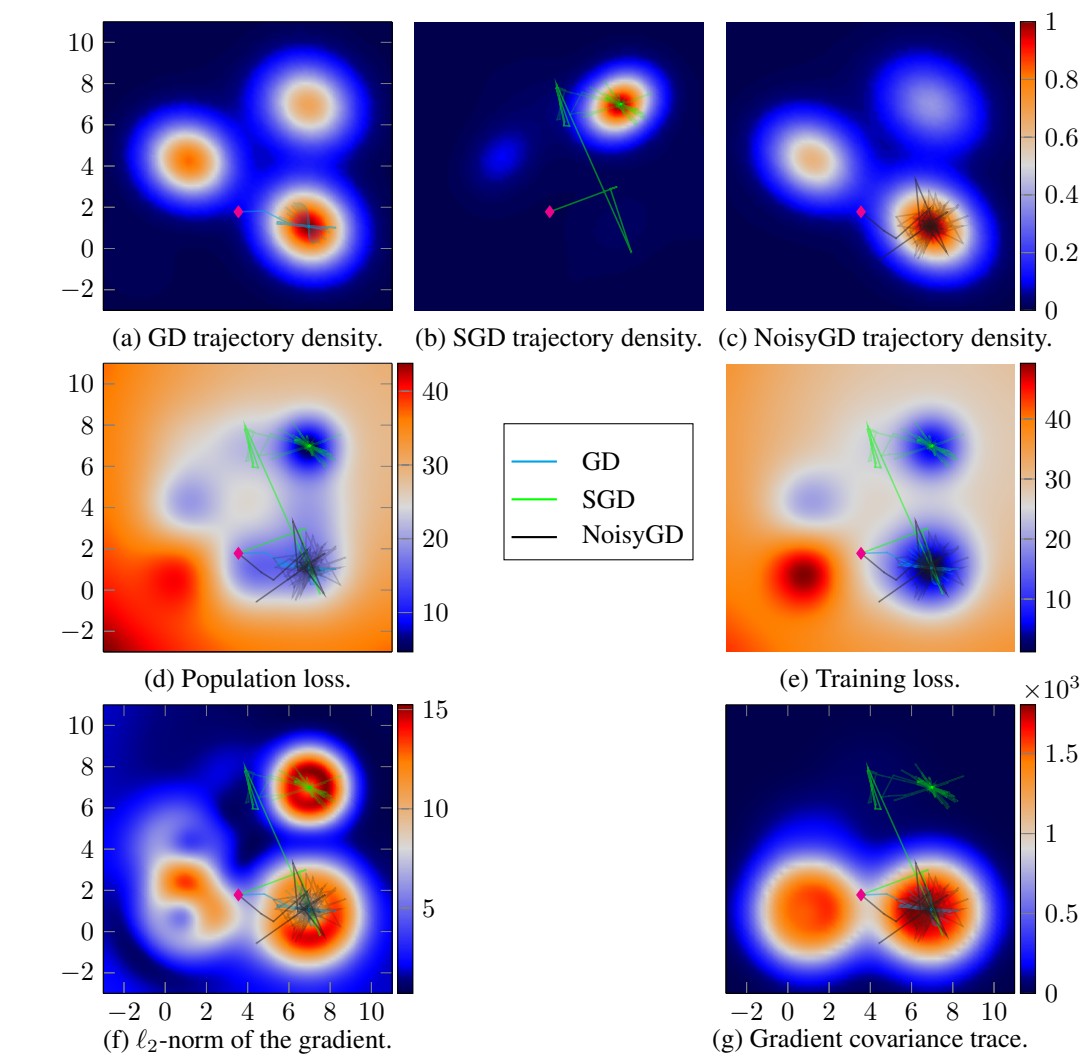

Figure 2: Heat maps of algorithm trajectory densities, population and training losses, gradient norms, and gradient covariance traces from the idealized experiment, with representative trajectories overlaid.

## 3.1 NOTATIONS

This paper focuses on supervised learning. A sample $z = (x, y)$ consists of an input $x \in \mathcal{X} = \mathbb{R}^d$ and a target $y \in \mathcal{Y} = \mathbb{R}$. Let $S = \{z_1, z_2, \ldots, z_N\}$ be a training set of size $N$, where the $z_i$ are i.i.d. samples from the population distribution $D$ on $\mathcal{Z} = \mathcal{X} \times \mathcal{Y}$. $L(z; \theta)$ denotes the loss function evaluated on sample $z$ at model parameters $\theta$. We slightly abuse these notations by writing the average loss on the training set $S$ as $L(S; \theta) = \frac{1}{N} \sum_{i=1}^{N} L(z_i; \theta)$ and the expected loss on the population distribution $D$ as $L(D; \theta) = \mathbb{E}_{z \sim D} [L(z; \theta)]$.

For a training set $S$, let $A_t(S)$ denote the solution obtained by applying SGD to $S$ for $t$ iterations, starting from initialization $A_0$. A specific SGD instantiation $A_T$ can be represented by $\{j_1, j_2, \ldots, j_T\}$, where $j_t$ indicates that sample $z_{j_t}$ in $S$ is selected at iteration $t$. $\mathbb{E}_{A_T} [f]$ takes the expectation of function $f$ over all possible $A_T$, given a fixed model initialization $A_0$ and a learning rate schedule $\{\eta_1, \eta_2, \ldots, \eta_T\}$. We can construct a perturbed training set for $S$ by replacing the $i$-th sample with a new one drawn from the population distribution: $S^i = \{z_1, \ldots, z_{i-1}, z'_i, z_{i+1}, \ldots, z_N\}, z'_i \sim D$. Unless stated otherwise, $\mathbb{E}_{z'_i} [f]$ denotes the expectation of function $f$ over $z'_i$ drawn from the population distribution $D$. For brevity, we define $J(v) = vv^T$ for any vector $v$.

## 3.2 DECOMPOSITION OF THE EXPECTED GENERALIZATION GAP

We derive a decomposition of the expected generalization gap under the following assumptions.

**Assumption 1.** *For $S \in \mathcal{Z}^N$, with probability $1 - \delta_{1,T}$, SGD obtains a solution whose batch gradient $\ell_2$-norm is bounded by $\epsilon_{1,T}$ after $T$ iterations, i.e.,*

$$\Pr\left(\|\nabla L\left(S; A_T\left(S\right)\right)\|_2 > \epsilon_{1,T}\right) < \delta_{1,T}$$

*for some $0 < \epsilon_{1,T}, \delta_{1,T} \ll 1$.*

Multiple studies have shown that overparameterized models can interpolate training sets under suitable conditions (Richtárik & Takác, 2020; Vaswani et al., 2019; Loizou et al., 2021). Consequently, Assumption 1 holds broadly across many machine learning problems.

**Assumption 2.** *For $S \in \mathcal{Z}^N$, with probability $1 - \delta_{2,T}$, the $\ell_2$-norm of the deviation between the solutions obtained by running SGD for $T$ iterations on $S$ and its perturbed training set $S^i$ is bounded by $\epsilon_{2,T}$, i.e.,*

$$\Pr\left(\left\|A_T\left(S\right) - A_T\left(S^i\right)\right\|_2 > \epsilon_{2,T}\right) < \delta_{2,T}$$

*for some $0 < \epsilon_{2,T}, \delta_{2,T} \ll 1$.*

Assumption 2 concerns the solution stability under single-sample replacements and holds when the training set is sufficiently large that such replacements have a small effect.

**Lemma 1.** *Consider a loss function $L$ whose value is bounded by $U_L$, with batch gradient $\ell_2$-norm bounded by $U_G$ and all third-order partial derivatives bounded by $U_J$. Assume the parameters are bounded as $\|\theta\|_2 \leq U_F$. If Assumptions 1 and 2 hold for $L$, the expected generalization gap satisfies*

$$\mathbb{E}_{S,A_T}\left[L\left(D; A_T\left(S\right)\right) - L\left(S; A_T\left(S\right)\right)\right] \tag{1}$$

$$= \frac{1}{N}\sum_{i=1}^{N}\mathbb{E}_{S,z_i',A_T}\left[L\left(z_i'; A_T\left(S\right)\right)\right] - \frac{1}{N}\sum_{i=1}^{N}\mathbb{E}_{S,z_i',A_T}\left[L\left(z_i'; A_T\left(S^i\right)\right)\right] \tag{2}$$

$$= \mathbb{E}_{S,A_T}\left[\frac{1}{2}\operatorname{Tr}\left(\nabla^2 L\left(S; A_T\left(S\right)\right)\frac{1}{N}\sum_{i=1}^{N}\mathbb{E}_{z_i'}\left[J\left(A_T\left(S^i\right) - A_T\left(S\right)\right)\right]\right)\right] \tag{3}$$

$$+ \mathcal{O}\left(\epsilon_{1,T}\epsilon_{2,T} + \delta_{1,T}\epsilon_{2,T}U_G + \delta_{1,T}\delta_{2,T}U_G U_F + \epsilon_{2,T}^3 U_J + \delta_{2,T}U_L\right). \tag{4}$$

Lemma 1 provides the expected generalization gap decomposition, with its proof given in Appendix B.1. $\mathbb{E}_{S,z_i',A_T}\left[L\left(z_i'; A_T\left(S\right)\right)\right]$ denotes the expectation of $L\left(z_i'; A_T\left(S\right)\right)$ over $S \sim D^N, z_i' \sim D$ and SGD instantiations initialized at $A_0$. The big-O term arises from the remainder of the Taylor expansion, and its constant does not depend on the problem setting. For the second-order Taylor expansion to be accurate, it suffices that all the third-order partial derivatives in a neighborhood of the solution are bounded by the smallest eigenvalue of its Hessian, scaled by $\epsilon_{2,T}$. This condition may fail for degenerate solutions, but note that flat directions contribute far less to the expected generalization gap than sharp ones. Combined with the alignment between the Hessian and the gradient covariance of SGD (Wu et al., 2022), this justifies the approximation.

The decomposition depends on the solution's Hessian and $\frac{1}{N}\sum_{i=1}^{N}\mathbb{E}_{z_i'}\left[J\left(A_T\left(S^i\right) - A_T\left(S\right)\right)\right]$. We denote this latter term as the *algorithmic variability*, which measures the sensitivity of the solution to single-sample replacements in the training set. Next, we will show that SGD automatically estimates and regularizes this variability term.

## 3.3 BOOTSTRAP ESTIMATION OF THE ALGORITHMIC VARIABILITY

We proceed to demonstrate that SGD uses the accumulated gradient covariance as a bootstrap estimate of part of a bound on the algorithmic variability. The analysis in this subsection relies on the data-dependent gradient noise of SGD and therefore does not extend to GD.

**Lemma 2.** *Consider the case where the model is trained with SGD on the training set $S$ for $M$ epochs, with each sample appearing exactly once in every epoch. Assume that*

*1. The learning rates are small, i.e., letting $Q = \max_t \eta_t$, we have $Q \ll 1$.*

2. *The operator norm of $\nabla^2 L\left(S; \theta\right)$ is uniformly bounded by a constant $C \ll \frac{1}{Q}$.*

*Then, the algorithmic variability can be bounded as*

$$\text{Tr}\left(\nabla^2 L\left(S, A_T\left(S\right)\right) \frac{1}{N} \sum_{i=1}^{N} \mathbb{E}_{z_i'}\left[J\left(A_T\left(S^i\right) - A_T\left(S\right)\right)\right]\right) \tag{5}$$

$$\leq \text{Tr}\left(\nabla^2 L\left(S, A_T\left(S\right)\right) \sum_{t=1}^{T} M\eta_t^2 \mathbb{E}_{z_i'}\left[J\left(\nabla L\left(z_i'; A_{t-1}\left(S\right)\right) - \nabla L\left(D; A_{t-1}\left(S\right)\right)\right)\right]\right) \tag{6}$$

$$+ \text{Tr}\left(\nabla^2 L\left(S, A_T\left(S\right)\right) \sum_{t=1}^{T} M\eta_t^2 J\left(\nabla L\left(S_{j_t}; A_{t-1}\left(S\right)\right) - \nabla L\left(D; A_{t-1}\left(S\right)\right)\right)\right) \tag{7}$$

$$+ \mathcal{O}\left(TQ\epsilon_{2,T}\left(\epsilon_{2,T}^3 U_J + \delta_{2,T} U_L U_F^3 + CQU_F\right) + T^2 Q^2 \left(\epsilon_{2,T}^3 U_J + \delta_{2,T} U_L U_F^3 + CQU_F\right)^2\right). \tag{8}$$

The proof of Lemma 2 is in Appendix B.2. It relies on the positive semi-definiteness of the solution's Hessian. This condition is guaranteed since SGD avoids solutions with negative Hessian eigenvalues almost surely (Mertikopoulos et al., 2020). Since $Q \ll 1$, $TQ = \mathcal{O}\left(1\right)$ for finite $T$, hence the big-O term is bounded.

**Theorem 1.** *Denote by $\Sigma_B^S\left(\theta\right)$ the gradient covariance of mini-batches of size $B$ evaluated on dataset $S$ at $\theta$. If the conditions of Lemma 2 hold, $\theta$ lies within a compact set $\Theta$, and $\nabla L\left(z_i'; \theta\right)$ is continuous with respect to $\theta$ on $\Theta$, then as the training set size $N \to \infty$, the difference between the accumulated population gradient covariance and the accumulated gradient covariance of SGD converges to $0$ almost surely, i.e.,*

$$\sum_{t=1}^{T} \mathbb{E}_{z_i'}\left[J\left(\nabla L\left(z_i'; A_{t-1}\left(S\right)\right) - \nabla L\left(D; A_{t-1}\left(S\right)\right)\right)\right] - \sum_{t=1}^{T} B\Sigma_B^S\left(A_{t-1}\left(S\right)\right) \overset{a.s.}{\to} 0. \tag{9}$$

Theorem 1 is the core contribution of this work, and its proof is given in Appendix B.3. Suppose SGD were to draw $K$ mini-batches of size $B$ from $S$ at $A_{t-1}\left(S\right)$, the empirical gradient covariance of these mini-batches would act as a bootstrap estimate of $\Sigma_B^S\left(A_{t-1}\left(S\right)\right)$ and converge to it as $K \to \infty$. Furthermore, $B\Sigma_B^S\left(A_{t-1}\left(S\right)\right)$ serves as an estimate of the population gradient covariance $\mathbb{E}_{z_i'}\left[J\left(\nabla L\left(z_i'; A_{t-1}\left(S\right)\right) - \nabla L\left(D; A_{t-1}\left(S\right)\right)\right)\right]$ and converges to it as $N \to \infty$. Hence, we interpret SGD as using the accumulated mini-batch gradient covariance as a bootstrap estimation of the accumulated population gradient covariance, which constitutes the first component of the algorithmic variability bound in equation 6.

Although Theorem 1 is an asymptotic result, our experiments show that the accumulated gradient covariance of SGD is strongly correlated with the algorithmic variability even for moderate $N$. We refer to the eigenvectors corresponding to the largest eigenvalues of a matrix as its principal eigendirections. For the accumulated gradient covariance matrix to accurately estimate the accumulated population gradient covariance, the span of the sample gradients must capture most of the principal eigendirections of the population gradient covariance, which requires $N$ to be at least as large as the number of principal eigendirections of the population gradient covariance. In practice, real data often reside in a low-dimensional subspace, which explains why the estimation is accurate even for moderate $N$.

### 3.4 IMPLICIT REGULARIZER AND GENERALIZATION

We now show how SGD implicitly regularizes the first part of the algorithmic variability bound in equation equation 6, thereby enhancing generalization. Smith et al. (2021) show that when resampling mini-batches of size $B$ without replacement, SGD implicitly regularizes the mean squared Euclidean distance between the sample gradients and the batch gradient, $\Gamma\left(\theta\right) = \frac{1}{N} \sum_{i=1}^{N} \|\nabla L\left(z_i; \theta\right) - \nabla L\left(S; \theta\right)\|_2^2$, with implicit regularizer $\frac{N-B}{N-1} \frac{\Gamma(\theta)}{B}$. Analogously, when resampling with replacement from $S$, SGD implicitly regularizes $\frac{\Gamma(\theta)}{B}$. By algebraic manipulation, we

show that this quantity equals the trace of the mini-batch gradient covariance:

$$\text{Tr}\left(\Sigma_B^S\left(\theta\right)\right) = \text{Tr}\left(\frac{\sum_{i=1}^{N} J\left(\nabla L\left(z_i;\theta\right) - \nabla L\left(S;\theta\right)\right)}{BN}\right) = \frac{\sum_{i=1}^{N} \|\nabla L\left(z_i;\theta\right) - \nabla L\left(S;\theta\right)\|_2^2}{BN}.$$

(10)

This implicit regularizer of SGD reduces the trace of the gradient covariance during training, thereby controlling the algorithmic variability. Since the expected generalization gap depends on the algorithmic variability, this implicit regularizer enables SGD to generalize better.

The second component of the algorithmic variability bound in equation 7 is neither estimated nor regularized by SGD. Analogous to the bootstrap estimation in Theorem 1, we introduce a plug-in estimator of this term as an explicit regularizer. For set $S$, at iteration $t$, we define *regularizer 1* as

$$\text{Reg1} = \lambda_1 \frac{1}{N}\sum_{i=1}^{N} \|\nabla L\left(z_i; A_{t-1}\left(S\right)\right) - \nabla L\left(S; A_{t-1}\left(S\right)\right)\|_2^2$$

(11)

and *regularizer 2* as

$$\text{Reg2} = \lambda_2 \|\nabla L\left(S_{j_t}; A_{t-1}\left(S\right)\right) - \nabla L\left(S; A_{t-1}\left(S\right)\right)\|_2^2,$$

(12)

with $\lambda_1$ and $\lambda_2$ denoting their respective strengths. These two regularizers correspond to estimates of the two components of the algorithmic variability bound. We evaluate the impact of these regularizers on generalization with numerical experiments in the following sections.

### 3.5 EMPIRICAL VALIDATION

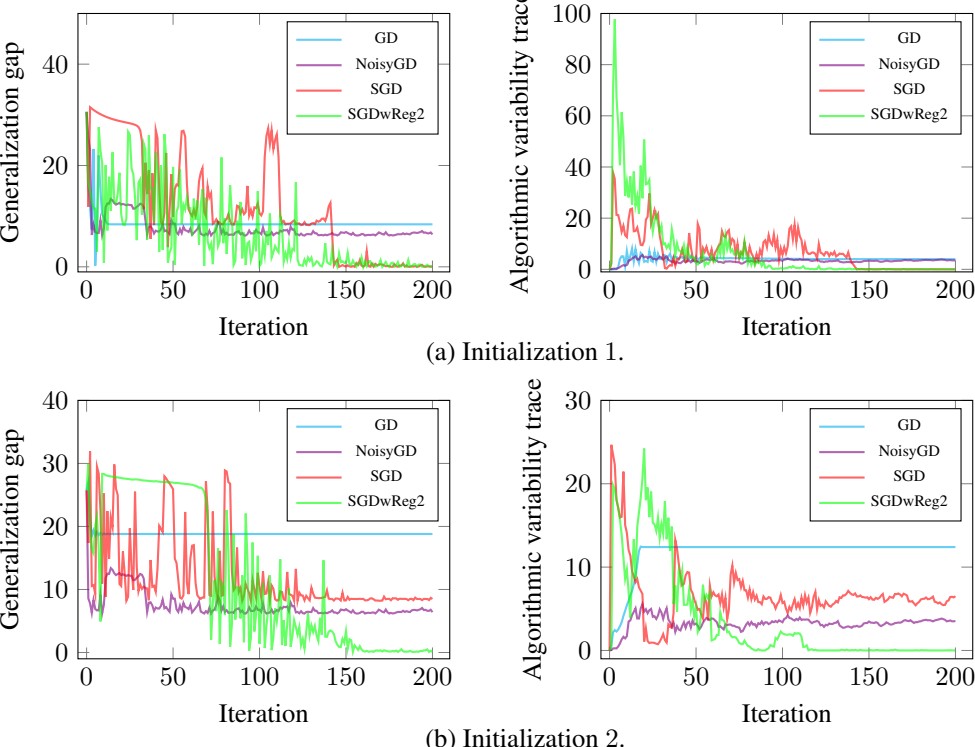

(a) Initialization 1.

(b) Initialization 2.

Figure 3: Trajectories of the generalization gap and the algorithmic variability trace versus iteration for GD, SGD, NoisyGD, and SGDwReg2, shown for two initializations in the idealized experiment.

We extend the idealized experiment in Section 2.4 to illustrate the relationship between algorithmic variability and generalization gap. In addition to the three algorithms considered above, we evaluate

SGDwReg2, which incorporates regularizer 2 into SGD. Figure 3 reports the trajectories of the generalization gap and the algorithmic variability trace under two different initializations. We observe that sharp decreases in the algorithmic variability trace coincide with reductions in the generalization gap, and algorithms ending with smaller algorithmic variability traces exhibit smaller generalization gaps. GD and NoisyGD have no control over the algorithmic variability, and we observe they end with larger variability trace and generalization gap in most experiments. We deliberately choose one case where SGD has poor generalization as Initialization 2. Notably, while SGD performs poorly under initialization 2, SGDwReg2 consistently reduces the algorithmic variability trace and achieves good generalization. These results align with our analysis above: while SGD only implicitly regularizes the first component of the algorithmic variability bound, incorporating regularizer 2 enables SGDwReg2 to regularize the full bound.

# 4 NUMERICAL EXPERIMENTS

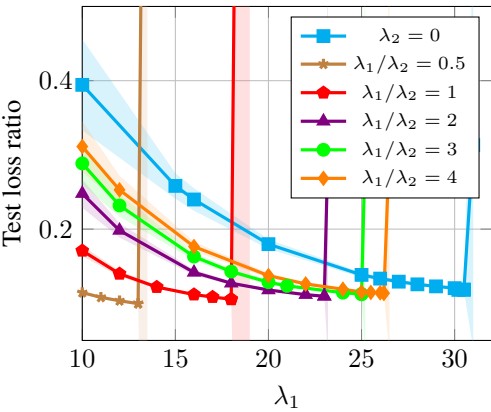

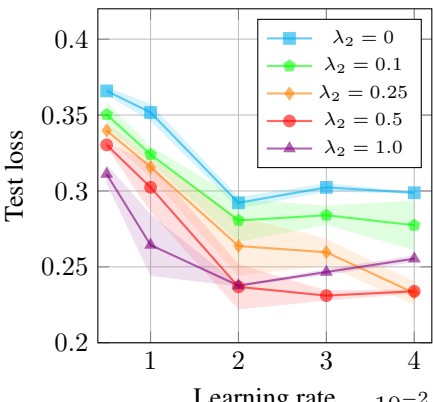

Figure 4: Average test loss ratios of SGD with regularizers 1 and 2 relative to the vanilla SGD benchmarks. Shaded areas represent standard deviations of the test loss ratios across different initializations.

Figure 5: Average test losses of SGD with regularizer 2. Shaded areas represent standard deviations across three runs with different random seeds.

## 4.1 SPARSE REGRESSION WITH DIAGONAL LINEAR NETWORKS

To evaluate the impact of regularizers 1 and 2 on generalization with moderately large training sets, we conduct experiments on sparse regression. We incorporate both regularizers 1 and 2 into SGD, for different fixed values of the relative strength ratio $\frac{\lambda_1}{\lambda_2}$, and compare their performance with that of the vanilla SGD benchmark. The model we use is the diagonal linear network (DLN), parameterized as $\theta = (\theta_a, \theta_b) \in \mathbb{R}^{2d}$. This model represents the function $f(x; \theta) = \langle \theta_a \odot \theta_b, x \rangle$, where $\odot$ denotes the element-wise multiplication. Despite its simplicity, the DLN is overparameterized and provides the non-convexity we seek (Pesme et al., 2021). We run the algorithms to minimize the mean squared error of the training sets, which consist of 40 samples. The detailed experimental setup is in Appendix A.2.

While different initializations strongly impact the test loss, the test loss ratios between different algorithms starting from the same initialization remain stable. Therefore, for each training set and initialization combination, we use the test loss of the vanilla SGD for that setting as the benchmark. The performance of each algorithm is evaluated by the ratios between its test loss and this benchmark.

Figure 4 shows that most test loss ratios fall below 1, indicating that incorporating explicit regularizers to SGD improves the average test loss of the solution. Within certain thresholds, increasing the regularization strength reduces the test loss. Notably, incorporating regularizer 2 can further reduce the test loss. Across the experimental settings, smaller relative strength ratios correspond to better generalization performance. Compared to the $\lambda_2 = 0$ curve, the setting $\frac{\lambda_1}{\lambda_2} = 0.5$ reduces the best test loss by 14%. These observations demonstrate that these regularizers improve generalization

for moderately large training sets. Moreover, consistent with our analysis in Section 3, while SGD only implicitly regularizes the first component of the algorithmic variability bound to help the model generalize, the best performance is achieved when both regularizers 1 and 2 are incorporated, thereby regularizing the full algorithmic variability bound.

## 4.2 DEEP NEURAL NETWORKS

To evaluate the effectiveness of the proposed regularizers in practical deep neural network training, we train a convolutional neural network (CNN) on the FashionMNIST dataset (Xiao et al., 2017). We use SGD with weight decay regularization as a benchmark. Owing to computational budget constraints, we incorporate only regularizer 2 and omit the batch gradient term in it. The detailed experimental setup is in Appendix A.3.

Figure 5 compares the performance of different regularization strengths and the benchmark, represented by the $\lambda_2 = 0$ curve. Overall, the average test loss decreases when we explicitly incorporate regularizer 2. For small regularization strengths, we observe consistent improvements over the benchmark. At larger regularization strengths, regularizer 2 yields better performance under small learning rates, but it also carries the risk of degrading test performance as the learning rate increases. These results demonstrate that an appropriately tuned regularizer 2 improves generalization.

## 4.3 COMPUTATIONAL OVERHEAD OF REGULARIZERS

One potential concern regarding the proposed regularizers is that both involve the batch gradient $\nabla L(S; A_{t-1}(S))$, and computing this term can incur large computation overhead. In practice, we can use approximation of the batch gradient when implementing the regularizers. For instance, the average of the previous $k$ mini-batch gradients can be used as an approximation to the batch gradient when the learning rate is not too large. In the experiment of Section 4.2, we omit the batch gradient term in regularizer 2 completely because the magnitude of the batch gradient becomes much smaller than that of most of the mini-batches quickly. In this case, the training time of SGDwReg is approximately 2.2 times of SGD.

## 5 RELATED WORK

**Solution sharpness perspective.** Many studies try to explain the generalization behavior of SGD from the perspective of solution sharpness. Keskar et al. (2016) show that the generalization drop of the model is caused by the sharp minimizer it converges to when using large batches. Yang et al. (2023); Wu & Su (2023) attribute the good generalization of a solution to its low sharpness. Moreover, Ma & Ying (2021); Wu et al. (2022); Ibayashi & Imaizumi (2021) show that stochasticity in SGD leads to solutions with low sharpness without explicit regularization.

However, these sharpness-based explanations suffer from the lack of invariance under reparameterization (Andriushchenko et al., 2023). Different parameterizations of the same function can yield drastically different sharpness values. This fact undermines the claim that the generalization performance of a function is directly correlated with its sharpness. Our perspective is related to the sharpness views in that the expected generalization gap decomposition involves the solution's Hessian, but crucially differs from them because it considers the entire training trajectory. Since reparameterization alters the training dynamics and can lead to different solutions, our perspective is not subject to invariance issues.

**Algorithmic stability perspective.** Another line of work connects generalization to algorithmic stability. Bousquet & Elisseeff (2002) and Elisseeff et al. (2005) define different kinds of stability and lay the foundation for this branch of work. Shalev-Shwartz et al. (2010) explore the connection between learnability and stability of empirical risk minimization. Recent works in this area include high-probability bounds (Feldman & Vondrak, 2019), hypothesis set stability (Foster et al., 2019), and uniformly stable algorithms (Bousquet et al., 2020). Regarding the generalization gap, Zhou et al. (2022) give a generalization gap bound based on the gradient variability on the training set, and Thomas et al. (2020) propose an estimation of the generalization gap based on the Hessian and gradient covariance at the solution evaluated on the population distribution.

Prior stability analyses typically yield worst-case generalization bounds under uniform smoothness assumptions, which can be overly conservative in highly non-convex settings. In contrast, our decomposition of the expected generalization gap is Hessian-weighted and evaluated at the solutions, thereby capturing the local curvature in regions of the loss landscape that the algorithm actually reaches. Free from uniform smoothness bounds, this framework enables us to isolate the impact of algorithmic variability and identify SGD's implicit regularization on the bootstrap estimate of algorithmic variability as the mechanism underlying its generalization advantage.

## 6 CONCLUSION

We provide an explanation of the generalization advantage of SGD based on a bootstrap estimation of the algorithmic variability. Specifically, we demonstrate that SGD implicitly regularizes the trace of the gradient covariance matrix, which serves as a bootstrap estimate of part of the algorithmic variability bound. This regularization guides SGD toward solutions that are robust to sampling noise, thereby enhancing generalization performance. While our theoretical analysis relies on specific assumptions on problem settings, numerical experiments in both synthetic and real-world settings show that our claims extend to broader settings. The experimental results demonstrate that incorporating the bootstrap estimates as explicit regularizers can effectively improve generalization in practice. These findings underscore the central role of the algorithmic variability in generalization and offer new insights into designing new regularizers to enhance generalization. An important open problem is whether the optimal regularization strength can be estimated from the training data or automatically tuned during training.

## REPRODUCIBILITY STATEMENT

The detailed experimental setups for the idealized experiment, the DLN experiment, and the CNN experiment are given in Appendix A.1, A.2, and A.3, respectively. Complete proofs for Lemma 1, Lemma 2, and Theorem 1 are given in Appendix B.1, B.2, and B.3. The source code for all experiments conducted in this work is included in the zipped supplementary materials.

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

## A   EXPERIMENTAL SETUP

### A.1   IDEALIZED EXPERIMENT

We consider a set of 30 different candidate functions as the population. Each function has a local minimum at $(7, 7)$, as well as an additional critical point. This second critical point is located at one point drawn uniformly from the set $\{(1, 1), (1, 4), (1, 7), (4, 1), (4, 4), (4, 7), (7, 1), (7, 4)\}$. With probability $\rho$, the additional critical point is a local maximum; otherwise, it is a local minimum. We set $\rho = 0.35$. Each critical point is modeled by a Gaussian function, with its height and width drawn from Gaussian distributions. We construct 10 training sets. Each training set contains 30 candidate functions, sampled with replacement from the population. We run all experiments for 200 iterations with an initial learning rate of $0.4$ and decay rate of $0.99$. Each algorithm is evaluated over 100 different initializations, arranged in a $10 \times 10$ grid on the domain $[0, 8] \times [0, 8]$, and the results are averaged.

This experiment was conducted on an Apple MacBook Pro equipped with an M1 Pro processor and 16 GB of memory. The typical running time for a single training set is approximately 19 minutes.

### A.2   SPARSE REGRESSION WITH DIAGONAL LINEAR NETWORKS

For each training set and initialization combination, we run experiments with three different model initializations and four training sets and average the results to account for the stochasticity. Each training set contains 40 samples whose inputs are drawn from a 100-d Gaussian distribution $\mathcal{N}(0, I_{100})$. The label for each sample is generated by taking the inner product between the true solution vector $\beta$ and the input, then adding a Gaussian noise to it:

$$y_i = \langle \beta, x_i \rangle + \xi, \xi \sim \mathcal{N}(0, 1). \tag{13}$$

The sparse true solution vector $\beta$ has 5 non-zero entries, randomly generated from a Gaussian distribution $\mathcal{N}(0, 2I_5)$.

For each combination of initialization and training set, we run the algorithms 4 times and take the average over the results. We use a constant learning rate of $0.01$ throughout the 200 training epochs.

Experiments were conducted using NVIDIA V100 GPUs with 32 GB of memory. For both one and two explicit regularizers, the typical running time of SGD is approximately 3.2 hours for 200 epochs at a given regularization strength.

### A.3   DEEP NEURAL NETWORKS

We conduct experiments with only regularizer 2, because it is much more tractable to compute than regularizer 1. We omit the batch gradient term $\nabla L(S; A_{t-1}(S))$ to further reduce the computational cost.

The CNN has two convolutional layers, whose structures are (in_channels=1, out_channels=32, kernel_size=3, stride=1, padding=1) and (in_channels=32, out_channels=64, kernel_size=3, stride=1, padding=1), and one fully-connected hidden layer with 128 nodes. We run each algorithm for 400 epochs with gradient clipping and batch size 32. To accelerate convergence, we let the learning rate decay by $1\%$ after each epoch. For the weight decay benchmark, we conduct a grid search over candidate values of the decay rate and select $0.01$ as the optimal setting.

Experiments were conducted using NVIDIA V100 GPUs with 32 GB of memory. The typical running times for the original SGD and SGD with the explicit regularizer are 2.5 and 5.5 hours for 400 epochs at a given regularization strength.

## B   PROOFS

### B.1   PROOF OF LEMMA 1

**Lemma 1.** *Consider a loss function $L$ whose value is bounded by $U_L$, with batch gradient $\ell_2$-norm bounded by $U_G$ and all third-order partial derivatives bounded by $U_J$. Assume the parameters are*

*bounded as $\|\theta\|_2 \leq U_F$. If Assumptions 1 and 2 hold for L, the expected generalization gap satisfies*

$$\mathbb{E}_{S,A_T} \left[ L \left( D; A_T \left( S \right) \right) - L \left( S; A_T \left( S \right) \right) \right] \tag{1}$$

$$= \frac{1}{N} \sum_{i=1}^{N} \mathbb{E}_{S,z_i',A_T} \left[ L \left( z_i'; A_T \left( S \right) \right) \right] - \frac{1}{N} \sum_{i=1}^{N} \mathbb{E}_{S,z_i',A_T} \left[ L \left( z_i'; A_T \left( S^i \right) \right) \right] \tag{2}$$

$$= \mathbb{E}_{S,A_T} \left[ \frac{1}{2} \text{Tr} \left( \nabla^2 L \left( S; A_T \left( S \right) \right) \frac{1}{N} \sum_{i=1}^{N} \mathbb{E}_{z_i'} \left[ J \left( A_T \left( S^i \right) - A_T \left( S \right) \right) \right] \right) \right] \tag{3}$$

$$+ \mathcal{O} \left( \epsilon_{1,T} \epsilon_{2,T} + \delta_{1,T} \epsilon_{2,T} U_G + \delta_{1,T} \delta_{2,T} U_G U_F + \epsilon_{2,T}^3 U_J + \delta_{2,T} U_L \right). \tag{4}$$

*Proof.* As mentioned in the main article, $\mathbb{E}_{z_i'} \left[ f \right]$ means $\mathbb{E}_{z_i' \sim D} \left[ f \right]$ and $\mathbb{E}_S \left[ f \right]$ means $\mathbb{E}_{S \sim D^N} \left[ f \right]$.

Note that for a specific realization of SGD, it is no longer symmetric in each sample of the training set. Intuitively, it makes a bigger difference when the replacement happens earlier rather than later. So, we will average over all the $N$ locations in the training set. We write the expected training loss over $S$ and $A_T$ in terms of sample losses:

$$\mathbb{E}_{S,A_T} \left[ L \left( S; A_T \left( S \right) \right) \right] = \mathbb{E}_{A_T} \left[ \mathbb{E}_S \left[ L \left( S; A_T \left( S \right) \right) \right] \right] \tag{14}$$

$$= \mathbb{E}_{A_T} \left[ \mathbb{E}_S \left[ \frac{1}{N} \sum_{i=1}^{N} L \left( z_i; A_T \left( S \right) \right) \right] \right] \tag{15}$$

$$= \mathbb{E}_{A_T} \left[ \frac{1}{N} \sum_{i=1}^{N} \mathbb{E}_S \left[ L \left( z_i; A_T \left( S \right) \right) \right] \right] \tag{16}$$

$$= \frac{1}{N} \sum_{i=1}^{N} \mathbb{E}_{S,A_T} \left[ L \left( z_i; A_T \left( S \right) \right) \right]. \tag{17}$$

Note that for a certain $i \in [N]$, $\mathbb{E}_S \left[ L \left( z_i; A_T \left( S \right) \right) \right] = \mathbb{E}_{S,z_i'} \left[ L \left( z_i'; A_T \left( S^i \right) \right) \right]$ and thus,

$$\mathbb{E}_{S,A_T} \left[ L \left( S; A_T \left( S \right) \right) \right] = \frac{1}{N} \sum_{i=1}^{N} \mathbb{E}_{S,z_i',A_T} \left[ L \left( z_i'; A_T \left( S^i \right) \right) \right]. \tag{18}$$

The expected generalization gap can be formulated as

$$\mathbb{E}_{S,A_T} \left[ L \left( D; A_T \left( S \right) \right) - L \left( S; A_T \left( S \right) \right) \right] \tag{19}$$

$$= \frac{1}{N} \sum_{i=1}^{N} \mathbb{E}_{S,z_i',A_T} \left[ L \left( z_i'; A_T \left( S \right) \right) \right] - \frac{1}{N} \sum_{i=1}^{N} \mathbb{E}_{S,z_i',A_T} \left[ L \left( z_i'; A_T \left( S^i \right) \right) \right] \tag{20}$$

$$= \frac{1}{N} \sum_{i=1}^{N} \mathbb{E}_{S,z_i',A_T} \left[ L \left( z_i'; A_T \left( S \right) \right) - L \left( z_i'; A_T \left( S^i \right) \right) \right]. \tag{21}$$

We apply a second-order Taylor expansion to the expression in equation 21:

$$\frac{1}{N} \sum_{i=1}^{N} \mathbb{E}_{S,z_i',A_T} \left[ L \left( z_i'; A_T \left( S \right) \right) - L \left( z_i'; A_T \left( S^i \right) \right) \right] \tag{22}$$

$$= \frac{1}{N} \sum_{i=1}^{N} \mathbb{E}_{S,z_i',A_T} \left[ \nabla L \left( z_i'; A_T \left( S^i \right) \right) \left( A_T \left( S \right) - A_T \left( S^i \right) \right) \right. \tag{23}$$

$$\left. + \frac{1}{2} \left( A_T \left( S \right) - A_T \left( S^i \right) \right)^T \nabla^2 L \left( z_i'; A_T \left( S^i \right) \right) \left( A_T \left( S \right) - A_T \left( S^i \right) \right) \right] \tag{24}$$

$$+ \mathcal{O} \left( \epsilon_{2,T}^3 U_J + \delta_{2,T} U_L \right). \tag{25}$$

The terms $\epsilon_{2,T}^3 U_J$ and $\delta_{2,T} U_L$ constitute the remainder of the second-order Taylor expansion.

Recall that Assumption 1 assumes that the gradient $\ell_2$-norm, $\left\| \nabla L \left( z_i'; A_T \left( S^i \right) \right) \right\|_2$, is bounded by $\epsilon_{1,T}$ with probability $1 - \delta_{1,T}$. Therefore, we can bound the first-order term in equation 22 by

$$\frac{1}{N} \sum_{i=1}^{N} \mathbb{E}_{S, z_i', A_T} \left[ \nabla L \left( z_i'; A_T \left( S^i \right) \right) \left( A_T \left( S \right) - A_T \left( S^i \right) \right) \right. \tag{26}$$

$$+ \frac{1}{2} \left( A_T \left( S \right) - A_T \left( S^i \right) \right)^T \nabla^2 L \left( z_i'; A_T \left( S^i \right) \right) \left( A_T \left( S \right) - A_T \left( S^i \right) \right) \right] \tag{27}$$

$$+ \mathcal{O} \left( \epsilon_{2,T}^3 U_J + \delta_{2,T} U_L \right) \tag{28}$$

$$= \frac{1}{N} \sum_{i=1}^{N} \mathbb{E}_{S, z_i', A_T} \left[ \frac{1}{2} \left( A_T \left( S \right) - A_T \left( S^i \right) \right)^T \nabla^2 L \left( z_i'; A_T \left( S^i \right) \right) \left( A_T \left( S \right) - A_T \left( S^i \right) \right) \right] \tag{29}$$

$$+ \mathcal{O} \left( \epsilon_{1,T} \epsilon_{2,T} + \delta_{1,T} \epsilon_{2,T} U_G + \delta_{1,T} \delta_{2,T} U_G U_F + \epsilon_{2,T}^3 U_J + \delta_{2,T} U_L \right) \tag{30}$$

$$= \frac{1}{N} \sum_{i=1}^{N} \mathbb{E}_{S, z_i', A_T} \left[ \frac{1}{2} \operatorname{Tr} \left( \nabla^2 L \left( z_i'; A_T \left( S^i \right) \right) J \left( A_T \left( S \right) - A_T \left( S^i \right) \right) \right) \right] \tag{31}$$

$$+ \mathcal{O} \left( \epsilon_{1,T} \epsilon_{2,T} + \delta_{1,T} \epsilon_{2,T} U_G + \delta_{1,T} \delta_{2,T} U_G U_F + \epsilon_{2,T}^3 U_J + \delta_{2,T} U_L \right). \tag{32}$$

The term $\epsilon_{1,T} \epsilon_{2,T}$ corresponds to the case where both Assumption 1 and 2 hold, the term $\delta_{1,T} \epsilon_{2,T} U_G$ corresponds to the case where Assumption 1 does not hold but Assumption 2 holds, and the term $\delta_{1,T} \delta_{2,T} U_G U_F$ corresponds to the case where neither of the two assumptions holds. The term corresponding to the case where Assumption 1 holds but Assumption 2 does not hold is dominated by $\delta_{2,T} U_L$.

Because sampling $\left( S, z_i' \right)$ and $\left( S^i, z_i \right)$ are symmetric,

$$\mathbb{E}_{S, z_i', A_T} \left[ f \left( S, S^i, z_i, z_i', A_T \right) \right] = \mathbb{E}_{S^i, z_i, A_T} \left[ f \left( S, S^i, z_i, z_i', A_T \right) \right]$$

for any function $f$. Therefore, we can exchange $z_i$ with $z_i'$ and $S$ with $S^i$ in equation 31, and then apply the bound on the solution deviation in Assumption 2 to obtain

$$\frac{1}{N} \sum_{i=1}^{N} \mathbb{E}_{S, z_i', A_T} \left[ \frac{1}{2} \operatorname{Tr} \left( \nabla^2 L \left( z_i'; A_T \left( S^i \right) \right) J \left( A_T \left( S \right) - A_T \left( S^i \right) \right) \right) \right] \tag{33}$$

$$= \frac{1}{N} \sum_{i=1}^{N} \mathbb{E}_{S^i, z_i, A_T} \left[ \frac{1}{2} \operatorname{Tr} \left( \nabla^2 L \left( z_i'; A_T \left( S^i \right) \right) J \left( A_T \left( S \right) - A_T \left( S^i \right) \right) \right) \right] \tag{34}$$

$$= \frac{1}{N} \sum_{i=1}^{N} \mathbb{E}_{S, z_i', A_T} \left[ \frac{1}{2} \operatorname{Tr} \left( \nabla^2 L \left( z_i; A_T \left( S \right) \right) J \left( A_T \left( S^i \right) - A_T \left( S \right) \right) \right) \right] \tag{35}$$

$$= \frac{1}{N} \sum_{i=1}^{N} \mathbb{E}_{S, A_T} \left[ \frac{1}{2} \operatorname{Tr} \left( \nabla^2 L \left( z_i; A_T \left( S \right) \right) \frac{1}{N} \sum_{q=1}^{N} \mathbb{E}_{z_q'} \left[ J \left( A_T \left( S^q \right) - A_T \left( S \right) \right) \right] \right) \right] \tag{36}$$

$$+ \mathcal{O}(\epsilon_{2,T}^3 U_J + \delta_{2,T} U_L) \tag{37}$$

$$= \mathbb{E}_{S, z_i', A_T} \left[ \frac{1}{2} \operatorname{Tr} \left( \frac{1}{N} \sum_{i=1}^{N} \nabla^2 L \left( z_i; A_T \left( S \right) \right) \frac{1}{N} \sum_{q=1}^{N} \mathbb{E}_{z_q'} \left[ J \left( A_T \left( S^i \right) - A_T \left( S \right) \right) \right] \right) \right] \tag{38}$$

$$+ \mathcal{O}(\epsilon_{2,T}^3 U_J + \delta_{2,T} U_L) \tag{39}$$

$$= \mathbb{E}_{S, A_T} \left[ \frac{1}{2} \operatorname{Tr} \left( \nabla^2 L \left( S; A_T \left( S \right) \right) \frac{1}{N} \sum_{i=1}^{N} \mathbb{E}_{z_i'} \left[ J \left( A_T \left( S^i \right) - A_T \left( S \right) \right) \right] \right) \right] \tag{40}$$

$$+ \mathcal{O}(\epsilon_{2,T}^3 U_J + \delta_{2,T} U_L). \tag{41}$$

Plugging equation 40-equation 41 into equation 31 gives the desired result:

$$\mathbb{E}_{S,A_T}\left[L\left(D;A_T\left(S\right)\right)-L\left(S;A_T\left(S\right)\right)\right] \tag{42}$$

$$= \mathbb{E}_{S,A_T}\left[\frac{1}{2}\operatorname{Tr}\left(\nabla^2 L\left(S;A_T\left(S\right)\right)\frac{1}{N}\sum_{i=1}^{N}\mathbb{E}_{z'_i}\left[J\left(A_T\left(S^i\right)-A_T\left(S\right)\right)\right]\right)\right] \tag{43}$$

$$+\mathcal{O}\left(\epsilon_{1,T}\epsilon_{2,T}+\delta_{1,T}\epsilon_{2,T}U_G+\delta_{1,T}\delta_{2,T}U_G U_F+\epsilon_{2,T}^3 U_J+\delta_{2,T}U_L\right). \tag{44}$$

$$\square$$

### B.2 PROOF OF LEMMA 2

**Lemma 2.** *Consider the case where the model is trained with SGD on the training set $S$ for $M$ epochs, with each sample appearing exactly once in every epoch. Assume that*

*1. The learning rates are small, i.e., letting $Q = \max_t \eta_t$, we have $Q \ll 1$.*

*2. The operator norm of $\nabla^2 L\left(S;\theta\right)$ is uniformly bounded by a constant $C \ll \frac{1}{Q}$.*

*Then, the algorithmic variability can be bounded as*

$$\operatorname{Tr}\left(\nabla^2 L\left(S,A_T\left(S\right)\right)\frac{1}{N}\sum_{i=1}^{N}\mathbb{E}_{z'_i}\left[J\left(A_T\left(S^i\right)-A_T\left(S\right)\right)\right]\right) \tag{5}$$

$$\leq \operatorname{Tr}\left(\nabla^2 L\left(S,A_T\left(S\right)\right)\sum_{t=1}^{T}M\eta_t^2\mathbb{E}_{z'_i}\left[J\left(\nabla L\left(z'_i;A_{t-1}\left(S\right)\right)-\nabla L\left(D;A_{t-1}\left(S\right)\right)\right)\right]\right) \tag{6}$$

$$+\operatorname{Tr}\left(\nabla^2 L\left(S,A_T\left(S\right)\right)\sum_{t=1}^{T}M\eta_t^2 J\left(\nabla L\left(S_{j_t};A_{t-1}\left(S\right)\right)-\nabla L\left(D;A_{t-1}\left(S\right)\right)\right)\right) \tag{7}$$

$$+\mathcal{O}\left(TQ\epsilon_{2,T}\left(\epsilon_{2,T}^3 U_J+\delta_{2,T}U_L U_F^3+CQU_F\right)+T^2 Q^2\left(\epsilon_{2,T}^3 U_J+\delta_{2,T}U_L U_F^3+CQU_F\right)^2\right). \tag{8}$$

*Proof.* For simplicity of exposition, and without loss of generality, we only consider the case of SGD with batch size 1 in this proof.

From a fixed model initialization $A_0$, the update rule of SGD leads to

$$A_T\left(S\right) = A_0 - \sum_{t=1}^{T}\eta_t\nabla L\left(S_{j_t};A_{t-1}\left(S\right)\right) \tag{45}$$

and

$$A_T\left(S^i\right) = A_0 - \sum_{t=1}^{T}\eta_t\nabla L\left(S_{j_t}^i;A_{t-1}\left(S^i\right)\right). \tag{46}$$

Thus,

$$A_T\left(S^i\right)-A_T\left(S\right) = \sum_{t=1}^{T}\eta_t\left(\nabla L\left(S_{j_t};A_{t-1}\left(S\right)\right)-\nabla L\left(S_{j_t}^i;A_{t-1}\left(S^i\right)\right)\right). \tag{47}$$

Now we prove by induction that

$$A_k\left(S^i\right)-A_k\left(S\right) = \sum_{t=1}^{k}\eta_t\left(\nabla L\left(S_{j_t};A_{t-1}\left(S\right)\right)-\nabla L\left(S_{j_t}^i;A_{t-1}\left(S\right)\right)\right) \tag{48}$$

$$+\mathcal{O}\left(kQ\left(\epsilon_{2,T}^3 U_J+\delta_{2,T}U_L U_F^3+CQU_F\right)\right), \tag{49}$$

for $k = [T]$.

**Base case:** Because $A_0(S) = A_0(S^i) = A_0$, it is easy to check that equation 48-equation 49 hold when $k = 1$.

**Inductive hypothesis:** Suppose equation 48-equation 49 hold for all $1 \leq k \leq p$.

**Inductive step:**

$$A_{p+1}(S^i) - A_{p+1}(S) \tag{50}$$

$$= A_p(S^i) - A_p(S) - \eta_{p+1} \nabla L\left(S^i_{j_{p+1}}; A_p(S^i)\right) + \eta_{p+1} \nabla L\left(S_{j_{p+1}}; A_p(S)\right) \tag{51}$$

$$= A_p(S^i) - A_p(S) + \eta_{p+1}\left(\nabla L\left(S_{j_{p+1}}; A_p(S)\right) - \nabla L\left(S^i_{j_{p+1}}; A_p(S)\right)\right) \tag{52}$$

$$- \eta_{p+1}\left(\nabla L\left(S^i_{j_{p+1}}; A_p(S^i)\right) - \nabla L\left(S^i_{j_{p+1}}; A_p(S)\right)\right). \tag{53}$$

The solution stability bound grows with the number of iterations (Hardt et al., 2016). Consequently, Assumption 2 holds with $\epsilon_{2,T}$ and $\delta_{2,T}$ for the entire training process. We can apply a Taylor expansion to the term in equation 53:

$$\nabla L\left(S^i_{j_{p+1}}; A_p(S^i)\right) - \nabla L\left(S^i_{j_{p+1}}; A_p(S)\right) \tag{54}$$

$$= \nabla^2 L\left(S^i_{j_{p+1}}; A_p(S)\right)\left(A_p(S^i) - A_p(S)\right) + \mathcal{O}\left(\epsilon^3_{2,T} U_J + \delta_{2,T} U_L U_F^3\right). \tag{55}$$

Recall that the Hessian operator norm is bounded by $C \ll \frac{1}{Q}$. Plugging equation 55 back into equation 53 obtains

$$A_{p+1}(S^i) - A_{p+1}(S) \tag{56}$$

$$= A_p(S^i) - A_p(S) + \eta_{p+1}\left(\nabla L\left(S_{j_{p+1}}; A_p(S)\right) - \nabla L\left(S^i_{j_{p+1}}; A_p(S)\right)\right) \tag{57}$$

$$- \eta_{p+1}\left(\nabla L\left(S^i_{j_{p+1}}; A_p(S^i)\right) - \nabla L\left(S^i_{j_{p+1}}; A_p(S)\right)\right) \tag{58}$$

$$= A_p(S^i) - A_p(S) + \eta_{p+1}\left(\nabla L\left(S_{j_{p+1}}; A_p(S)\right) - \nabla L\left(S^i_{j_{p+1}}; A_p(S)\right)\right) \tag{59}$$

$$- \eta_{p+1} \nabla^2 L\left(S^i_{j_{p+1}}; A_p(S)\right)\left(A_p(S^i) - A_p(S)\right) + \mathcal{O}\left(Q\left(\epsilon^3_{2,T} U_J + \delta_{2,T} U_L U_F^3\right)\right) \tag{60}$$

$$= A_p(S^i) - A_p(S) + \eta_{p+1}\left(\nabla L\left(S_{j_{p+1}}; A_p(S)\right) - \nabla L\left(S^i_{j_{p+1}}; A_p(S)\right)\right) \tag{61}$$

$$+ \mathcal{O}\left(Q\left(\epsilon^3_{2,T} U_J + \delta_{2,T} U_L U_F^3 + CQU_F\right)\right) \tag{62}$$

$$= \sum_{t=1}^{p} \eta_t\left(\nabla L\left(S_{j_t}; A_{t-1}(S)\right) - \nabla L\left(S^i_{j_t}; A_{t-1}(S)\right)\right) \tag{63}$$

$$+ \mathcal{O}\left(pQ\left(\epsilon^3_{2,T} U_J + \delta_{2,T} U_L U_F^3 + CQU_F\right)\right) \tag{64}$$

$$+ \eta_{p+1}\left(\nabla L\left(S_{j_{p+1}}; A_p(S)\right) - \nabla L\left(S^i_{j_{p+1}}; A_p(S)\right)\right) \tag{65}$$

$$+ \mathcal{O}\left(Q\left(\epsilon^3_{2,T} U_J + \delta_{2,T} U_L U_F^3 + CQU_F\right)\right) \tag{66}$$

$$= \sum_{t=1}^{p+1} \eta_t\left(\nabla L\left(S_{j_t}; A_{t-1}(S)\right) - \nabla L\left(S^i_{j_t}; A_{t-1}(S)\right)\right) \tag{67}$$

$$+ \mathcal{O}\left((p+1)Q\left(\epsilon^3_{2,T} U_J + \delta_{2,T} U_L U_F^3 + CQU_F\right)\right). \tag{68}$$

Thus, equation 48-equation 49 also hold for the case $k = p + 1$. By the principle of mathematical induction,

$$A_k(S^i) - A_k(S) = \sum_{t=1}^{k} \eta_t\left(\nabla L\left(S_{j_t}; A_{t-1}(S)\right) - \nabla L\left(S^i_{j_t}; A_{t-1}(S)\right)\right) \tag{69}$$

$$+ \mathcal{O}\left(kQ\left(\epsilon^3_{2,T} U_J + \delta_{2,T} U_L U_F^3 + CQU_F\right)\right) \tag{70}$$

for $k = [T]$.

For each $i \in [N]$, there are $M$ indices $t$ such that $j_t = i$. For simplicity of notation, we require that the sample sequence is not shuffled for different epochs, so that $j_m = j_{m+rN} = m, m \in [N], r \in [M-1]$. However, the result also holds when the sample sequence is shuffled in each epoch.

It follows that

$$\mathrm{Tr}\left(\nabla^2 L\left(S, A_T\left(S\right)\right) \frac{1}{N} \sum_{i=1}^{N} \mathbb{E}_{z_i'}\left[J\left(A_T\left(S^i\right) - A_T\left(S\right)\right)\right]\right) \tag{71}$$

$$= \mathrm{Tr}\left(\nabla^2 L\left(S, A_T\left(S\right)\right) \frac{1}{N} \sum_{i=1}^{N} \mathbb{E}_{z_i'}\left[J\left(\sum_{t=1}^{T} \eta_t\left(\nabla L\left(S_{j_t}; A_{t-1}\left(S\right)\right) - \nabla L\left(S_{j_t}^i; A_{t-1}\left(S\right)\right)\right)\right)\right]\right) \tag{72}$$

$$+ \mathcal{O}\left(TQ\epsilon_{2,T}\left(\epsilon_{2,T}^3 U_J + \delta_{2,T} U_L U_F^3 + CQ U_F\right) + T^2 Q^2 \left(\epsilon_{2,T}^3 U_J + \delta_{2,T} U_L U_F^3 + CQ U_F\right)^2\right). \tag{73}$$

Note that

$$\sum_{i=1}^{N} \mathbb{E}_{z_i'}\left[\left(\nabla L\left(S_{j_m}; A_{m-1}\left(S\right)\right) - \nabla L\left(S_{j_m}^i; A_{m-1}\left(S\right)\right)\right)\right. \tag{74}$$

$$\left.\left(\nabla L\left(S_{j_n}; A_{n-1}\left(S\right)\right) - \nabla L\left(S_{j_n}^i; A_{n-1}\left(S\right)\right)\right)^T\right] = 0 \tag{75}$$

when $j_m \neq j_n$. In this case, since $S$ and $S^i$ only differ in the $i$-th sample, either $S_{j_m} = S_{j_m}^i$ or $S_{j_n} = S_{j_n}^i$, making the outer product of the gradient differences zero. Hence,

$$\mathrm{Tr}\left(\nabla^2 L\left(S, A_T\left(S\right)\right) \frac{1}{N} \sum_{i=1}^{N} \mathbb{E}_{z_i'}\left[J\left(\sum_{t=1}^{T} \eta_t\left(\nabla L\left(S_{j_t}; A_{t-1}\left(S\right)\right) - \nabla L\left(S_{j_t}^i; A_{t-1}\left(S\right)\right)\right)\right)\right]\right) \tag{76}$$

$$= \mathrm{Tr}\left(\nabla^2 L\left(S, A_T\left(S\right)\right)\right. \tag{77}$$

$$\left.\sum_{i=1}^{N} \mathbb{E}_{z_i'}\left[J\left(\sum_{r=0}^{M-1} \eta_{i+rN}\left(\nabla L\left(S_{j_{i+rN}}; A_{i+rN-1}\left(S\right)\right) - \nabla L\left(S_{j_{i+rN}}^i; A_{i+rN-1}\left(S\right)\right)\right)\right)\right]\right) \tag{78}$$

$$\leq \mathrm{Tr}\left(\nabla^2 L\left(S, A_T\left(S\right)\right)\right. \tag{79}$$

$$\left.\sum_{i=1}^{N} \mathbb{E}_{z_i'}\left[\sum_{r=0}^{M-1} M\eta_{i+rN}^2 J\left(\nabla L\left(S_{j_{i+rN}}; A_{i+rN-1}\left(S\right)\right) - \nabla L\left(S_{j_{i+rN}}^i; A_{i+rN-1}\left(S\right)\right)\right)\right]\right) \tag{80}$$

$$= \mathrm{Tr}\left(\nabla^2 L\left(S, A_T\left(S\right)\right) \sum_{t=1}^{T} M\eta_t^2 \mathbb{E}_{z_{j_t}' \sim D}\left[J\left(\nabla L\left(S_{j_t}; A_{t-1}\left(S\right)\right) - \nabla L\left(S_{j_t}^{j_t}; A_{t-1}\left(S\right)\right)\right)\right]\right) \tag{81}$$

$$= \mathrm{Tr}\left(\nabla^2 L\left(S, A_T\left(S\right)\right) \sum_{t=1}^{T} M\eta_t^2 \mathbb{E}_{z_{j_t}' \sim D}\left[J\left(\nabla L\left(S_{j_t}; A_{t-1}\left(S\right)\right) - \nabla L\left(z_{j_t}'; A_{t-1}\left(S\right)\right)\right)\right]\right) \tag{82}$$

$$= \mathrm{Tr}\left(\nabla^2 L\left(S, A_T\left(S\right)\right) \sum_{t=1}^{T} M\eta_t^2 \mathbb{E}_{z_i'}\left[J\left(\nabla L\left(S_{j_t}; A_{t-1}\left(S\right)\right) - \nabla L\left(z_i'; A_{t-1}\left(S\right)\right)\right)\right]\right) \tag{83}$$

$$= \mathrm{Tr}\left(\nabla^2 L\left(S, A_T\left(S\right)\right) \mathbb{E}_{z_i'}\left[\sum_{t=1}^{T} M\eta_t^2 J\left(\nabla L\left(S_{j_t}; A_{t-1}\left(S\right)\right) - \nabla L\left(z_i'; A_{t-1}\left(S\right)\right)\right)\right]\right). \tag{84}$$

Equation 77-equation 78 aggregate all iterations which select the same samples in the $\sum_{r=0}^{M-1}$ sum, and the factor $\frac{1}{N}$ is absent because there is exactly one $i$ for which this expectation is non-zero. The inequality in equation 79-equation 80 results from the fact that, for any positive semi-definite matrix $G \in \mathbb{R}^{d \times d}$ and any vector sequence $\{u_p \in \mathbb{R}^d : p \in [M]\}$,

$$\text{Tr}\left(GJ\left(\sum_{p=1}^{M} u_p\right)\right) \leq \text{Tr}\left(GM \sum_{p=1}^{M} J(u_p)\right), \tag{85}$$

which can be derived as follows:

$$\text{Tr}\left(GM \sum_{p=1}^{M} J(u_p)\right) - \text{Tr}\left(GJ\left(\sum_{p=1}^{M} u_p\right)\right) \tag{86}$$

$$= M \sum_{p=1}^{M} \|u_p\|_G^2 - \left\|\sum_{p=1}^{M} u_p\right\|_G^2 \tag{87}$$

$$= \frac{1}{2} \sum_{p,q=1,p\neq q}^{M} \|u_p - u_q\|_G^2 \tag{88}$$

$$\geq 0. \tag{89}$$

By plugging the results in equation 76-equation 84 into equation 71-equation 73 and rearranging the terms, we obtain

$$\text{Tr}\left(\nabla^2 L(S, A_T(S)) \frac{1}{N} \sum_{i=1}^{N} \mathbb{E}_{z_i'}\left[J\left(A_T(S^i) - A_T(S)\right)\right]\right) \tag{90}$$

$$\leq \text{Tr}\left(\nabla^2 L(S, A_T(S)) \mathbb{E}_{z_i'}\left[\sum_{t=1}^{T} M\eta_t^2 J\left(\nabla L(S_{j_t}; A_{t-1}(S)) - \nabla L(z_i'; A_{t-1}(S))\right)\right]\right) \tag{91}$$

$$+ \mathcal{O}\left(TQ\epsilon_{2,T}\left(\epsilon_{2,T}^3 U_J + \delta_{2,T} U_L U_F^3 + CQU_F\right) + T^2 Q^2 \left(\epsilon_{2,T}^3 U_J + \delta_{2,T} U_L U_F^3 + CQU_F\right)^2\right) \tag{92}$$

$$= \text{Tr}\left(\nabla^2 L(S, A_T(S)) \mathbb{E}_{z_i'}\left[\sum_{t=1}^{T} M\eta_t^2 J\left(\left(\nabla L(S_{j_t}; A_{t-1}(S)) - \nabla L(D; A_{t-1}(S))\right)\right.\right.\right. \tag{93}$$

$$\left.\left.\left. + \left(\nabla L(D; A_{t-1}(S)) - \nabla L(z_i'; A_{t-1}(S))\right)\right)\right]\right) \tag{94}$$

$$+ \mathcal{O}\left(TQ\epsilon_{2,T}\left(\epsilon_{2,T}^3 U_J + \delta_{2,T} U_L U_F^3 + CQU_F\right) + T^2 Q^2 \left(\epsilon_{2,T}^3 U_J + \delta_{2,T} U_L U_F^3 + CQU_F\right)^2\right). \tag{95}$$

Note that

$$\mathbb{E}_{z_i'}\left[\left(\nabla L(S_{j_t}; A_{t-1}(S)) - \nabla L(D; A_{t-1}(S))\right)\left(\nabla L(D; A_{t-1}(S)) - \nabla L(z_i'; A_{t-1}(S))\right)^T\right] \tag{96}$$

$$= \left(\nabla L(S_{j_t}; A_{t-1}(S)) - \nabla L(D; A_{t-1}(S))\right) \mathbb{E}_{z_i'}\left[\left(\nabla L(D; A_{t-1}(S)) - \nabla L(z_i'; A_{t-1}(S))\right)^T\right] \tag{97}$$

$$= \left(\nabla L(S_{j_t}; A_{t-1}(S)) - \nabla L(D; A_{t-1}(S))\right)\left(\nabla L(D; A_{t-1}(S)) - \mathbb{E}_{z_i'}\left[\nabla L(z_i'; A_{t-1}(S))\right]\right)^T \tag{98}$$

$$= \left(\nabla L(S_{j_t}; A_{t-1}(S)) - \nabla L(D; A_{t-1}(S))\right)\left(\nabla L(D; A_{t-1}(S)) - \nabla L(D; A_{t-1}(S))\right)^T \tag{99}$$

$$= 0. \tag{100}$$

Plugging this identity back into equation 93-equation 94 yields the desired result:

$$\text{Tr}\left(\nabla^2 L\left(S, A_T\left(S\right)\right)\frac{1}{N}\sum_{i=1}^{N}\mathbb{E}_{z_i'}\left[J\left(A_T\left(S^i\right) - A_T\left(S\right)\right)\right]\right) \tag{101}$$

$$\leq \text{Tr}\left(\nabla^2 L\left(S, A_T\left(S\right)\right)\mathbb{E}_{z_i'}\left[\sum_{t=1}^{T}M\eta_t^2 J\Big(\left(\nabla L\left(S_{j_t}; A_{t-1}\left(S\right)\right) - \nabla L\left(D; A_{t-1}\left(S\right)\right)\right)\right.\right. \tag{102}$$

$$\left.\left.+ \left(\nabla L\left(D; A_{t-1}\left(S\right)\right) - \nabla L\left(z_i'; A_{t-1}\left(S\right)\right)\right)\Big)\right]\right) \tag{103}$$

$$+ \mathcal{O}\Big(TQ\epsilon_{2,T}\left(\epsilon_{2,T}^3 U_J + \delta_{2,T}U_L U_F^3 + CQU_F\right) + T^2 Q^2\left(\epsilon_{2,T}^3 U_J + \delta_{2,T}U_L U_F^3 + CQU_F\right)^2\Big) \tag{104}$$

$$= \text{Tr}\left(\nabla^2 L\left(S, A_T\left(S\right)\right)\sum_{t=1}^{T}M\eta_t^2\mathbb{E}_{z_i'}\left[J\left(\nabla L\left(D; A_{t-1}\left(S\right)\right) - \nabla L\left(z_i'; A_{t-1}\left(S\right)\right)\right)\right]\right) \tag{105}$$

$$+ \text{Tr}\left(\nabla^2 L\left(S, A_T\left(S\right)\right)\sum_{t=1}^{T}M\eta_t^2 J\left(\nabla L\left(S_{j_t}; A_{t-1}\left(S\right)\right) - \nabla L\left(D; A_{t-1}\left(S\right)\right)\right)\right) \tag{106}$$

$$+ \mathcal{O}\Big(TQ\epsilon_{2,T}\left(\epsilon_{2,T}^3 U_J + \delta_{2,T}U_L U_F^3 + CQU_F\right) + T^2 Q^2\left(\epsilon_{2,T}^3 U_J + \delta_{2,T}U_L U_F^3 + CQU_F\right)^2\Big). \tag{107}$$

$$\square$$

### B.3 Proof of Theorem 1

**Theorem 1.** *Denote by $\Sigma_B^S\left(\theta\right)$ the gradient covariance of mini-batches of size $B$ evaluated on dataset $S$ at $\theta$. If the conditions of Lemma 2 hold, $\theta$ lies within a compact set $\Theta$, and $\nabla L\left(z_i'; \theta\right)$ is continuous with respect to $\theta$ on $\Theta$, then as the training set size $N \to \infty$, the difference between the accumulated population gradient covariance and the accumulated gradient covariance of SGD converges to 0 almost surely, i.e.,*

$$\sum_{t=1}^{T}\mathbb{E}_{z_i'}\left[J\left(\nabla L\left(z_i'; A_{t-1}\left(S\right)\right) - \nabla L\left(D; A_{t-1}\left(S\right)\right)\right)\right] - \sum_{t=1}^{T}B\Sigma_B^S\left(A_{t-1}\left(S\right)\right) \overset{a.s.}{\to} 0. \tag{9}$$

*Proof.* We denote a training set of size $N$ as $S = \{z_1, z_2, \ldots, z_N\}, z_i \sim D, i \in [N]$. We use $z_i$ to denote samples in $S$, and $z'$ to denote an independent sample drawn from either the population distribution $D$ or the empirical distribution $D_{emp}^S$ associated with $S$. We proceed to define the variables

$$Z_i\left(\theta\right) = \nabla L\left(z_i; \theta\right) - \nabla L\left(D; \theta\right), \quad z_i \sim D, \quad i \in [N] \tag{108}$$

$$\Sigma^D\left(\theta\right) = \mathbb{E}_{z' \sim D}\left[J\left(\nabla L\left(z'; \theta\right) - \nabla L\left(D; \theta\right)\right)\right] \tag{109}$$

$$\bar{Z}_N\left(\theta\right) = \frac{1}{N}\sum_{i=1}^{N}Z_i\left(\theta\right). \tag{110}$$

Then, we can write

$$\mathbb{E}_{z' \sim D_{emp}^S} \left[ J \left( \nabla L \left( z'; \theta \right) - \nabla L \left( S; \theta \right) \right) \right] \tag{111}$$

$$= \mathbb{E}_{z' \sim D_{emp}^S} \left[ J \left( \nabla L \left( z'; \theta \right) - \nabla L \left( D; \theta \right) + \nabla L \left( D; \theta \right) - \nabla L \left( S; \theta \right) \right) \right] \tag{112}$$

$$= \mathbb{E}_{z' \sim D_{emp}^S} \left[ J \left( \left( \nabla L \left( z'; \theta \right) - \nabla L \left( D; \theta \right) \right) - \left( \nabla L \left( S; \theta \right) - \nabla L \left( D; \theta \right) \right) \right) \right] \tag{113}$$

$$= \mathbb{E}_{z' \sim D_{emp}^S} \left[ J \left( \nabla L \left( z'; \theta \right) - \nabla L \left( D; \theta \right) \right) \right] - \mathbb{E}_{z' \sim D_{emp}^S} \left[ J \left( \nabla L \left( D; \theta \right) - \nabla L \left( S; \theta \right) \right) \right] \tag{114}$$

$$= \frac{1}{N} \sum_{i=1}^{N} J \left( Z_i \left( \theta \right) \right) - J \left( \bar{Z}_N \left( \theta \right) \right). \tag{115}$$

Let $v^{(j)}$ indicate the $j$-th entry of vector $v$. Due to the continuity of $\nabla L \left( z'; \theta \right)$, $Z_i^{(j)} \left( \theta \right)$ is a continuous function of $\theta$ for any $z_i$. Also, from the bound on the $\ell_2$-norm of the gradient, $Z_i^{(j)} \left( \theta \right)$ is bounded by a function $h \left( z_i \right)$ for any $z_i$ and $\theta$, where $h$ is an integrable function of $z_i$ with respect to the distribution $D$. With these conditions, according to Theorem 2 in Jennrich (1969), for any $j, k \in [N]$,

$$\frac{1}{N} \sum_{i=1}^{N} Z_i^{(j)} \left( \theta \right) Z_i^{(k)} \left( \theta \right) \overset{a.s.}{\to} \mathbb{E}_{z_1 \sim D} \left[ Z_1^{(j)} \left( \theta \right) Z_1^{(k)} \left( \theta \right) \right] = \Sigma_{jk}^D \left( \theta \right) \tag{116}$$

uniformly for all $\theta \in \Theta$ as $N \to \infty$. Because $Z_i$ has a finite number of entries,

$$\frac{1}{N} \sum_{i=1}^{N} J \left( Z_i \left( \theta \right) \right) \overset{a.s.}{\to} \Sigma^D \left( \theta \right) \tag{117}$$

uniformly for all $\theta \in \Theta$ as $N \to \infty$.

From the strong law of large numbers, as the training set size approaches infinity, the mean of any gradient entry over the training set converges almost surely to its population mean. Since $\bar{Z}_N^{(j)} \left( \theta \right)$ represents the difference between the mean gradient over the training set $S$ and the population gradient, for any $j \in [N]$,

$$\bar{Z}_N^{(j)} \left( \theta \right) \overset{a.s.}{\to} 0 \tag{118}$$

uniformly for all $\theta \in \Theta$ as $N \to \infty$. Because $\bar{Z}_N$ has a finite number of entries,

$$J \left( \bar{Z}_N \left( \theta \right) \right) \overset{a.s.}{\to} 0 \tag{119}$$

uniformly for all $\theta \in \Theta$ as $N \to \infty$.

Combining equation 117 and equation 119 leads to

$$\mathbb{E}_{z' \sim D_{emp}^S} \left[ J \left( \nabla L \left( z'; \theta \right) - \nabla L \left( S; \theta \right) \right) \right] - \Sigma^D \left( \theta \right) \overset{a.s.}{\to} 0 \tag{120}$$

uniformly for any $\theta \in \Theta$ as $N \to \infty$.

At $\theta$, the gradient covariance of the mini-batches of size $B$ sampled from dataset $S$ can be expressed as the empirical gradient covariance $\Sigma_B^S \left( \theta \right) = \frac{1}{B} \mathbb{E}_{z' \sim D_{emp}^S} \left[ J \left( \nabla L \left( z'; \theta \right) - \nabla L \left( S; \theta \right) \right) \right]$. Thus, with the uniform convergence in equation 120, we obtain

$$\sum_{t=1}^{T} \mathbb{E}_{z' \sim D} \left[ J \left( \nabla L \left( z'; A_{t-1} \left( S \right) \right) - \nabla L \left( D; A_{t-1} \left( S \right) \right) \right) \right] - \sum_{t=1}^{T} B \Sigma_B^S \left( A_{t-1} \left( S \right) \right) \overset{a.s.}{\to} 0 \tag{121}$$

as $N \to \infty$, which is equivalent to the desired result. $\qquad \square$

