# OpenReview forum: "A Bootstrap Perspective on Stochastic Gradient Descent"
_ICLR.cc/2026/Conference — Submitted to ICLR 2026_

### Official Review · Reviewer_aZc8 · 2025-10-30

**Soundness:** 2
**Presentation:** 2
**Contribution:** 1
**Rating:** 2
**Confidence:** 5

**Summary:**

This paper presents a theoretical framework for understanding the generalization properties of Stochastic Gradient Descent (SGD). The authors decompose the generalization gap and introduce the concept of "algorithmic variability", which they analyze through the lens of statistical bootstrapping. Based on this decomposition, the authors construct two novel regularizers and empirically validate that their inclusion can lead to improved generalization performance on tasks including sparse regression and neural network training.
However, there are still some concerns to me. Therefore I lean to a rejection at the time being.
Specifically, I am not sure whether the idea in this paper has significant differences to algorithm stability, and whether the derivation of this paper is meaningful.
See below for more details.

**Strengths:**

1. The paper posits that SGD uses the gradient variability (caused by mini-batch sampling) as a "bootstrap estimate.
2. This paper proves that the expected generalization gap is determined by the trace of the product of the solution's Hessian matrix and the "algorithmic variability" matrix.
3. This paper designs a new regularizer based on the theoretical findings.
4. The authors further provide empirical evidence on this regularizer.

**Weaknesses:**

1. [Major Concern] It seems that Assumption 2 directly leads to a small Variability (Eqn 3). However, the authors did not discuss it much. If so, I cannot be convinced that Eqn (3) is the dominate term compared to Eqn (4), where Eqn (4) also contains the epsilon[2, T] term.
2. [Major Concern] I am not convinced that this paper has significant differences with the line of algorithmic stability. The authors claim in Line 466 that "this paper considers "Hessian-weighted and evaluated at the solutions"". It seems that algorithmic stability can include this case with pretty minor changes. Due to the simplicity, algorithm stability just bound the Hession with smoothness, and use iteration to reach the solution. But starting from the definition of algorithm stability, these are not necessary. The authors shold provide more evidence on how this paper performs differently with algorithm stability.

[Minor]
1. The authors claim that "we prove rigorously that by implicitly regularizing the trace of the gradient covariance matrix, SGD controls the algorithmic variability." According to the paper's derivation, the algorithmic variability is bounded by two components (corresponding to the latter term in Eq. 6 and Eq. 7). While the authors convincingly connect the implicit regularization of SGD, as identified by Smith et al. (2021), to the first component (Eq. 6), they do not provide evidence or argumentation that SGD also implicitly regularizes the second component (Eq. 7). Consequently, the claim that SGD "controls the algorithmic variability" in its entirety appears to be an overstatement.
This significantly limits their theoretical contribution, as the work seems to demonstrate that vanilla SGD only addresses a part of the problem identified by the authors.
2. The paper's analysis of the proposed regularizers, Reg1 and Reg2, lacks sufficient depth regarding their interplay and individual utility. For instance, given that the authors identify Reg1 as an existing *implicit* regularizer of SGD, a crucial discussion is missing on the utility of its *explicit* inclusion. What is the tangible difference between applying Reg1 explicitly versus relying on its implicit effect? Would applying only Reg2, which is the component not addressed by vanilla SGD, be a more practical and principled approach? The paper would be substantially strengthened by ablation studies that dissect the individual contributions of Reg1 and Reg2 and clarify their roles in guiding SGD towards better-generalizing solutions.
3. The practical significance of this work is severely hampered by the unaddressed computational overhead of the proposed regularizers. Both Reg1 and Reg2, as defined, require the computation of the full-batch gradient at each training step. This is a prohibitive cost for large-scale datasets and fundamentally contradicts the core philosophy of SGD, which is designed precisely to avoid such computations. The absence of any discussion on this issue, or on potential efficient approximations, makes it difficult to assess the empirical value of the proposed method. As it stands, the practical guidance offered by the paper appears limited.

**Questions:**

See above.

---

> ### Author Response · Authors · 2025-11-22
>
> We thank the reviewers for the detailed review and useful suggestions, which we will incorporate in the revised version.
>
> > It seems that Assumption 2 directly leads to a small Variability (Eqn 3). However, the authors did not discuss it much. If so, I cannot be convinced that Eqn (3) is the dominate term compared to Eqn (4), where Eqn (4) also contains the epsilon[2, T] term.
>
> Assumption 2 is crucial for bounding solution deviation and controlling the Taylor remainder, but it does not by itself lead to Eqn (3). Eqn (3) arises as the second-order term from the Taylor expansion, using both Assumptions 1 and 2 together with loss derivative bounds. You are also completely correct in that the terms in Eqn (4) involving epsilon[2,T] must be treated carefully. These terms can be controlled by having appropriate values for epsilon[1,T] and delta[1,T]. In practice, epsilon[1,T] and delta[1,T] are often orders of magnitude smaller than epsilon[2,T], owning to the ability of over-parameterized models to interpolate the training data.
>
> > I am not convinced that this paper has significant differences with the line of algorithmic stability.
>
> We agree with the reviewers that, with uniform smoothness, algorithmic stability can lead to bounds on the Hessian and thereby generalization gap. However, stability bounds are typically expressed in the $\ell_2$-norm, which is direction-insensitive. We suggest in Section 2.4 of the paper that the anisotropic, data-dependent gradient noise of SGD is what induces it to solutions with better generalization. Thus, rather than deriving a stability-based generalization gap bound, our goal is to recognize how the interaction between the solution Hessian and algorithmic variability affects generalization. Both the solution Hessian and algorithmic variability are highly anisotropic, and if we start from the definition of algorithm stability, it is hard to characterize the contribution of each dimension. Furthermore, the definition of algorithm stability needs information of the population. In contrast, our bootstrapping perspective provides a method to evaluate the stability solely from training trajectories and suggests that SGD does the evaluation implicitly.
>
> > the work seems to demonstrate that vanilla SGD only addresses a part of the problem identified by the authors.
>
> We thank the reviewers for pointing that out. Their understanding here is definitely correct, as the implicit regularizer of vanilla SGD regularizes the first component of the bound. We agree with them that it is more appropriate to clarify that SGD controls part of the algorithmic variability bound. We will change the text in the revision.
>
> > What is the tangible difference between applying Reg1 explicitly versus relying on its implicit effect? Would applying only Reg2, which is the component not addressed by vanilla SGD, be a more practical and principled approach? The paper would be substantially strengthened by ablation studies that dissect the individual contributions of Reg1 and Reg2 and clarify their roles in guiding SGD towards better-generalizing solutions.
>
> We thank the reviewers for the valuable suggestion. Currently, we show the effect of explicit including Reg1 in the DLN experiment (results shown in Section 4.1). The blue curve in Figure 4 corresponds to applying Reg1 explicitly, and we can see that, with appropriate regularization strength, it outperforms relying only on the implicit regularizer of SGD. As the reviewers suggested, applying only Reg2 is indeed more practical due to its lighter cost. We show its effect in the CNN experiment (results shown in Section 4.2). It can improve generalization with a moderate overhead. We will consider conducting ablation studies on a wider range of datasets and models in future work.
>
> > The practical significance of this work is severely hampered by the unaddressed computational overhead of the proposed regularizers. Both Reg1 and Reg2, as defined, require the computation of the full-batch gradient at each training step. This is a prohibitive cost for large-scale datasets and fundamentally contradicts the core philosophy of SGD, which is designed precisely to avoid such computations.
>
> We fully agree with the reviewers that the original core philosophy of SGD is to avoid loading the whole dataset in every training step. We propose Reg1 and Reg2 in their exact theoretical forms as derived from the bootstrapping view. In practice, they can be implemented with different levels of approximation. For instance, in our experiment in Section 4.2, we use $\lambda_2||\nabla L\left(S_{j_t};A_{t-1}\left(S\right)\right)||_2^2$ as an approximation, since after only a few learning steps, the batch gradient magnitude becomes much smaller than most of the individual sample gradients. We will add the computational overhead of the proposed regularizers in the revision.

---

### Official Review · Reviewer_TGMQ · 2025-10-31

**Soundness:** 2
**Presentation:** 2
**Contribution:** 1
**Rating:** 4
**Confidence:** 3

**Summary:**

The paper studies SGD's impact on generalization for machine learning models. Based on the provided  analyses, it proposes two regularization schemes, which are shown to benefit generalization for a few toy datasets.

**Strengths:**

The question raised in the paper is important and the paper tsts a new regularization method based on the analyses and shows that it might benefit generalization

**Weaknesses:**

The theoretical contribution appears to be incremental, as, to my understanding, the main insights came from Smith et al. (2021).  The empirical evaluation is very limited, as the results are tested only on a very specific synthetic dataset with a sparse prior and FashionMNIST.

**Questions:**

1) I did not understand how the analyses are specific to the SGD as opposed to the non-stochastic GD. As the opening sentence of the abstract mentions the difference between generalization of GD and SGD as a motivation, I would like to ask the authors to elaborate more on this. How can we see from the bounds derived in the paper that SGD might outperform GD?

2) As for the regularizers part, what are the novel insights made in the paper compared to Smith et al. (2021)?

---

> ### Author Response · Authors · 2025-11-22
>
> We sincerely appreciate the reviewers’ careful reading and constructive feedback.
>
> > I did not understand how the analyses are specific to the SGD as opposed to the non-stochastic GD. As the opening sentence of the abstract mentions the difference between generalization of GD and SGD as a motivation, I would like to ask the authors to elaborate more on this. How can we see from the bounds derived in the paper that SGD might outperform GD?
>
> The implicit bound on Eqn(6) is specific to SGD, as it arises from the sampling noise of SGD. In contrast, GD has no control over this term. In Section 3.5, we show with experimental results that GD has larger algorithmic variability than SGD. Since the average generalization gap is dependent on the algorithimc variability, SGD will outperform GD. We will make this comparison clearer in the revision.
>
> > As for the regularizers part, what are the novel insights made in the paper compared to Smith et al. (2021)?
>
> Smith et al. (2021) characterize the implicit regularizer of SGD (corresponding to Reg1) and suggest that it penalizes “non-uniform” regions where the minibatch gradients vary greatly. We employ a bootstrapping view to establish a concrete link between the implicit regularizer in their work and the average generalization gap, showing that the implicit regularizer improves the generalization gap through controlling part of the algorithmic variability bound. Furthermore, our bootstrapping view naturally leads to an additional regularizer (Reg2 in Lines 327-336). This additional regularizer complements the implicit regularizer in their work, and jointly they bound the full algorithmic variability. As we show with experimental results in Section 4, explicitly incorporating both Reg1 and Reg2 obtains the best generalization performance.

---

### Official Review · Reviewer_7CRg · 2025-11-01

**Soundness:** 2
**Presentation:** 2
**Contribution:** 2
**Rating:** 2
**Confidence:** 2

**Summary:**

The paper tries to understand SGD from the view of bootstrapping: SGD favors minima with smaller variance of stochastic gradient.

**Strengths:**

1. The top example in Section 2 is attractive and illustrative.

**Weaknesses:**

1. The presentation of the theoretical part is a bit confusing.
- The theoretical results are listed as Lemmas 1 and 2 as well as Proposition 1, without a theorem that usually serves as the center of discussions. This makes me confused about what is the main theoretical contribution of the paper.
- The discussions after Lemmas 1 and 2 mainly discuss why the lemmas hold, and do not actually help with the understanding of the theoretical results (especially for Lemma 2, whose righthand side has a lot of terms).
2. My understanding is that the core of the theoretical analysis is the correspondence of Equations (6) and (7) with Equations (10) and (11), which provides a viewpoint from the implicit regularization of SGD by "bootstrapping" the gradients. However, this part lacks a comparison against GD or noisy GD.
3. According to my understanding, the technical contribution is minor. Lemmas 1 and 2 are basically Taylor expansion, and Proposition 1 is basically the strong law of large numbers.

I would honestly confess that I do not understand all the details of the paper, and would be happy to discuss with the authors, other reviewers and the AC. My score of 2 currently represents my unconfident understanding. i think the intuition of the paper is good, but the theoretical part may need improvements.

**Questions:**

1. Can the authors show more details of the algorithm SGDwReg2, especially how to estimate the term Reg2?
- If Reg2 is estimated in an exact way, then SGDwReg2 requires knowledge of the entire dataset at each minibatch update. In this case, is it possible to design an adaptation of SGD that incorporates the idea of SGDwReg2 but without the requirement of the entire dataset?
- If Reg2 is approximated, can the authors show the details of approximation?
2. How does the bootstrapping view compare with the idea of variance-reduction techniques like SVRG?

---

> ### Author Response · Authors · 2025-11-22
>
> Thank you for your valuable comments. We acknowledge that some technical details may require further exposition. We would be very happy to discuss any unclear points with you, other reviewers, and the AC.
>
> > The theoretical results are listed as Lemmas 1 and 2 as well as Proposition 1, without a theorem that usually serves as the center of discussions. This makes me confused about what is the main theoretical contribution of the paper.
>
> We intend Proposition 1 to be the core theoretical contribution of this work. Our main theoretical contribution of this paper is that SGD uses the accumulated gradient covariance as a bootstrap estimate of part of the algorithmic variability bound and implicitly regularizes it to enhance generalization. We will revise this part to better emphasize its importance and make our main contribution clear.
>
> > My understanding is that the core of the theoretical analysis is the correspondence of Equations (6) and (7) with Equations (10) and (11), which provides a viewpoint from the implicit regularization of SGD by "bootstrapping" the gradients. However, this part lacks a comparison against GD or noisy GD.
>
> Your understanding is correct. The core is the correspondence of Eqn (6) to Eqn (10) (or, equivalently, Eqn (11)), which captures the implicit regularizer of SGD arising from the gradient noise, and also Eqn (7) to Eqn (12). Since the data-dependent gradient noise is unique to SGD, neither GD nor NoisyGD is aware of the information in the gradient covariance, and therefore unable to control these two terms. In Section 3.5, we also show with the experiment results that GD and NoisyGD have larger algorithmic variability compared to SGD. We will revise the text to emphasize that this implicit regularizer is unique to SGD.
>
> > Can the authors show more details of the algorithm SGDwReg2, especially how to estimate the term Reg2?
>
> Sure. SGDwReg2 is simply using Reg2 as an explicit regularizer and applying SGD to optimize the resulting regularized objective function. The expression of Reg2 in Eqn (12) is its exact form. In practice, one way to approximate the batch gradient is to use a moving average of the previous k gradients, as the learning rate is required to be small. In our experiment in Section 4.2, we use $\lambda_2||\nabla L\left(S_{j_t};A_{t-1}\left(S\right)\right)||_2^2$ as an approximation, since we observe that the batch gradient has a much smaller norm than most of the individual sample gradient after a few learning steps.
>
> > How does the bootstrapping view compare with the idea of variance-reduction techniques like SVRG?
>
> Thank you for the insightful comparison. Variance-reduction techniques like SVRG are designed to *actively* regularize the gradient variance. In comparison, out goal of introducing the bootstrapping view is to characterize how SGD estimate the algorithmic variability with the accumulated gradient covariance and thus control the variability *without* explicit regularization.

---

> > ### Comment · Reviewer_7CRg · 2025-11-23
> >
> > Many thanks for the clarifications! I now have a better understanding of the paper, but would still hesitate to rate it to be above the acceptance threshold.
> >
> > The main reason I feel that the reasoning is inadequate is that the regularizers Reg1 and Reg2 seem to be characterized with experiment results only. Without theoretical justifications, this claim is not clear because the quantitative characterization of how the regularization works is lacking. An illustrating example of on a simple example (like the one used in Section 2) would suffice.
> >
> > It would also be better if the authors could implement the changes that they promised to make. Updating the PDF is allowed during the discussion.

---

> > > ### Author Response · Authors · 2025-12-02
> > >
> > > Thank you for engaging in the discussion and taking effort to understand our work. We want to provide the following explanation:
> > >
> > > Regarding the theoretical origin of the two regularizers, Reg1 is a direct result of Theorem 1 (the previous Proposition 1), since it penalizes exactly the trace of the gradient covariance of SGD. Reg2 derives from the decomposition of the algorithmic variability bound in Lemma 2, accounting for the remaining term (Eqn (7)) not regularized by Reg1.
> > >
> > > Again, we are grateful to you and all the reviewers for your valuable feedback. We have submitted a revised version incorporating your suggestions.

---

### Official Review · Reviewer_1WA5 · 2025-11-01

**Soundness:** 2
**Presentation:** 3
**Contribution:** 1
**Rating:** 2
**Confidence:** 4

**Summary:**

This paper aims to provide a novel eccplanation for the superior genetalozation property of SGD compared wirh GD, from a boostrap perspctive. Specifically, under certein assumptions, the authors show that the generalization error can be decomposed into a dominant Hessian=-preconditioned algorithmic variability term and several small terms. They further argue that the algorithmic variavbilit is stronhly correlated to the accumulated empirical covriance of gradients. As a consequence, they empirically estalish that SGD regularizes algorithmic variability as a bootstrap estimate, and hence improving the generalization error through this correlation.

**Strengths:**

This paper is clearly written and has a nice structure.

**Weaknesses:**

Although the authors provide an upper bound on the generalization error via algorithmic stability, the paper does not explicitly establish how SGD regularizes this term theoretically. Moreover, there is no theoretical characterization of the generalization gap between SGD and GD. Another concern arises from the assumptions: while Assumption 1 appears standard, Assumption 2 is rather demanding and may not hold in many scenarios: existing theoretical results generally suggest that the upper bound on uniform algorithmic stability grows with the number of iterations. This implies that the bias term, rather than variance, often dominates the generalization error. From this perspective, the argument that “SGD generalizes better because it regularizes the gradient variance” may not be entirely convincing.

**Questions:**

No further questions.

**Details Of Ethics Concerns:**

This work is purely theoretical and has no negative ethical concerns.

---

> ### Author Response · Authors · 2025-11-22
>
> Thank you for your time and careful review of our paper.
>
> > Although the authors provide an upper bound on the generalization error via algorithmic stability, the paper does not explicitly establish how SGD regularizes this term theoretically.
>
> In Proposition 1, we derive that the accumulated gradient covariance of SGD is a bootstrap estimate of part of the algorithmic variability bound (corresponding to Eqn (6)). Then, in Section 3.4, we cite [1] to show that SGD implicitly regularizes the trace of the gradient covariance during training. Combining the above arguments together, we conclude that "This implicit regularizer of SGD reduces the trace of the gradient covariance during training, thereby controlling the algorithmic variability" (Line 324-325).
>
> > Moreover, there is no theoretical characterization of the generalization gap between SGD and GD.
>
> In this work, we define the generalization gap as the difference between the training and test loss for the solution of an algorithm. Among SGD and GD, only SGD implicitly regularizes the algorithmic variability to enhance generalization. We will emphasize that GD has no control over the algorithmic variability in the revision.
>
> > Another concern arises from the assumptions: while Assumption 1 appears standard, Assumption 2 is rather demanding and may not hold in many scenarios: existing theoretical results generally suggest that the upper bound on uniform algorithmic stability grows with the number of iterations. This implies that the bias term, rather than variance, often dominates the generalization error. From this perspective, the argument that “SGD generalizes better because it regularizes the gradient variance” may not be entirely convincing.
>
> We introduce Assumption 2 to investigate the dominant term in the average generalization gap. The intuition is that when the size of the training dataset is large, replacing one sample with a new one would only lead to small change in the solution. We would also like to note that algorithm stability bounds do not necessarily scale with the number of iterations. For instance, [2] proves that for $\beta$-smooth and $L$-Lipschitz function $f$, $E[||w_T-w_T'||]\leq \frac{2L}{N}\sum_{t=1}^T\alpha_t$, where $w_T$ and $w_T'$ are solutions obtained by SGD on training sets differing in one sample, $T$ is the number of learning steps, $N$ is the size of the training sets, and $\alpha_t$ is the step size at step $t$. If we use an exponential decay schedule for $\{\alpha_t\}$, then $\sum_{t=1}^T\alpha_t$ is bounded with a bound that does not depend on $T$, leading to $E[||w_T-w_T'||]=\mathcal{O}(\frac{L}{N})$.
>
> As to the dominance of the bias term in the generalization error, we have to admit we are a bit confused with how it can be implied from the algorithmic stability's dependence on the number of iterations. Could you please further elaborate on that point?
>
>
> [1] Smith, S.L., Dherin, B., Barrett, D.G. and De, S., 2021. On the origin of implicit regularization in stochastic gradient descent. arXiv preprint arXiv:2101.12176.
>
> [2] Moritz Hardt, Benjamin Recht, and Yoram Singer. Train faster, generalize better: Stability of stochastic gradient descent. In proceeding of ICML, pages 1225–1234, 2016.

---

### Meta-Review · Area_Chair_WuE1 · 2025-12-25

**Summary:**

This paper aims to provide a novel explanation for the superior generalization property of SGD compared with GD, from a bootstrap perspective. the reviewers commented various concerns such as: 1)he theoretical contribution appears to be incremental, as, to my understanding, the main insights came from Smith et al. (2021). 2) The empirical evaluation is very limited, as the results are tested only on a very specific synthetic dataset with a sparse prior and FashionMNIST 3)  the paper does not explicitly establish how SGD regularizes this term theoretically. Moreover, there is no theoretical characterization of the generalization gap between SGD and GD.

Although the authors provided the feedbacks, it seems to me that they are not convincing to the reviewers and their opinions remain the same. Based on these reasons, I can not recommend the acceptance.

**Reviewer Concerns:**

As mentioned in the meta review.

**Reviewer Scores:**

after reading the reviewers' comments, i do not think that the score would be changed if the reviwers participate fully in the discussion. One reviewer (Reviewer 7CRg) mentioned that the feedback from the authors is not convincing enough to change the score of acceptance threshold.

---

### Decision · Program_Chairs · 2026-01-26

Reject